# Phylogenetically and catabolically diverse diazotrophs reside in deep-sea cold seep sediments

Xiyang Dong [1,2,3,9] ✉, Chuwen Zhang[1,2,9], Yongyi Peng [1,2], Hong-Xi Zhang[4,5], Ling-Dong Shi [6], Guangshan Wei[1], Casey R. J. Hubert [7], Yong Wang [4,5] ✉ & Chris Greening [8]

Microbially mediated nitrogen cycling in carbon-dominated cold seep environments remains poorly understood. So far anaerobic methanotrophic archaea (ANME-2) and their sulfate-reducing bacterial partners (SEEP-SRB1 clade) have been identified as diazotrophs in deep sea cold seep sediments. However, it is unclear whether other microbial groups can perform nitrogen fixation in such ecosystems. To fill this gap, we analyzed 61 metagenomes, 1428 metagenome-assembled genomes, and six metatranscriptomes derived from 11 globally distributed cold seeps. These sediments contain phylogenetically diverse nitrogenase genes corresponding to an expanded diversity of diazotrophic lineages. Diverse catabolic pathways were predicted to provide ATP for nitrogen fixation, suggesting diazotrophy in cold seeps is not necessarily associated with sulfate-dependent anaerobic oxidation of methane. Nitrogen fixation genes among various diazotrophic groups in cold seeps were inferred to be genetically mobile and subject to purifying selection. Our findings extend the capacity for diazotrophy to five candidate phyla (Altarchaeia, Omnitrophota, FCPU426, Caldatribacteriota and UBA6262), and suggest that cold seep diazotrophs might contribute substantially to the global nitrogen balance.

Cold seeps occur in continental margins worldwide. At these sites, there is discharge of biologically or geologically sourced hydrocarbons, ranging in complexity from methane to the varying constituents of petroleum[1,2]. Cold seeps are often classified as slow-flow mineral-prone or high-flux mud-prone systems according to their hydrocarbon fluid regime[1]. They span oil and gas seeps, methane seeps, gas hydrates, asphalt volcanoes, mud volcanoes, brine pools, and brine basins among others. The seeping hydrocarbons support the development of extensive local diversity of archaea and bacteria, dominated by aerobic methane-oxidizing bacteria (MOB, e.g., members of the methanotrophic family Methylococcaceae) mainly at the oxygen-rich sediment-water interface[3] and microbial consortia of anaerobic methane-oxidizing archaea (ANME) with sulfate-reducing bacteria (SRB) within anoxic sediment layers[4–6]. Various studies have

[1]Key Laboratory of Marine Genetic Resources, Third Institute of Oceanography, Ministry of Natural Resources, Xiamen, China. [2]School of Marine Sciences, Sun Yat-Sen University, Zhuhai, China. [3]Southern Marine Science and Engineering Guangdong Laboratory (Zhuhai), Zhuhai, China. [4]Institute for Marine Engineering, Shenzhen International Graduate School, Tsinghua University, University Town, Shenzhen, China. [5]Department of Life Science, Institute of Deep-sea Science and Engineering, Chinese Academy of Sciences, Sanya, China. [6]College of Environmental and Resource Sciences, Zhejiang University, Hangzhou, China. [7]Department of Biological Sciences, University of Calgary, Calgary, AB, Canada. [8]Department of Microbiology, Biomedicine Discovery Institute, Clayton, VIC, Australia. [9]These authors contributed equally: Xiyang Dong, Chuwen Zhang. ✉e-mail: dongxiang@tio.org.cn; wangyong@sz.tsinghua.edu.cn

revealed microorganisms that oxidize non-methane hydrocarbons, such as ethane, butane, propane, liquid alkanes and aromatic hydrocarbons, also inhabit these environments[7–12]. In contrast with the rich and expanding knowledge of microbial hydrocarbon oxidation at cold seeps, little is known about how microbiomes in these ecosystems control the cycling of other essential nutrients. Seeping hydrocarbons introduce little nitrogen into these carbon-dominated systems, making cold seep sediments inherently limited by nitrogen supply to support biomass production[1,13].

Biological nitrogen fixation (diazotrophy)—the reduction of atmospheric dinitrogen gas ($N_2$) to ammonia ($NH_3$) with concomitant hydrogen gas ($H_2$) production—is a critical source of bioavailable nitrogen for living organisms[14–16]. The key enzymes mediating this process are nitrogenases, which include three forms distinguished by their active site metal cofactors: molybdenum-iron nitrogenase Nif (Mo-Fe), vanadium-iron nitrogenase Vnf (V-Fe) and iron-only nitrogenase Anf (Fe-Fe)[17,18]. All three nitrogenase forms are structurally and functionally similar, each containing two protein components: a dinitrogenase reductase (NifH, VnfH, or AnfH) and a catalytic component (NifDK, VnfDGK, or AnfDGK). Most biological $N_2$ fixation is catalyzed by the more efficient Mo-Fe nitrogenase, while Fe-V and Fe-Fe nitrogenases are alternative enzymes used in Mo-limited settings[19,20]. A combination of rate measurements, lab cultivation, flow cytometry, molecular analysis, and cellular imaging have revealed that diazotrophs are active throughout the oceans. Multiple cyanobacterial diazotrophs are responsible for a substantial portion of new nitrogen input in the surface ocean[21–24]. Various diazotrophs are also active in both surface and deeper waters, including diverse heterotrophic Proteobacteria[25–28]. Over the past decade, the deep benthos has been found to host diverse groups of previously unrecognized diazotrophs that actively and significantly contribute to local nitrogen balance, including members of Acidobacteria, Firmicutes, Nitrospirae, Gammaproteobacteria and Deltaproteobacteria[15,29,30]. Despite these advances, there remains limited knowledge about the distribution and evolution of biological nitrogen fixation in sediments from the deep sea, which covers nearly two-thirds of the Earth.

Multiple lines of evidence have demonstrated diazotrophy in different deep-sea cold seep sediments. These include a methane seep in the South China Sea, a mud volcano offshore Costa Rica, gas hydrate mounds in the Gulf of Mexico, and an active methane seep in the Eel River Basin[13,31–34]. Based on $^{15}N_2$ tracer experiments coupled with nanoSIMS, to date only two cold seep taxa have been identified as diazotrophs, the methanotrophic ANME-2 archaea and their sulfate-reducing bacterial partners of the SEEP-SRB1 clade[31,32,34,35]. However, PCR amplicon surveys targeting the nitrogenase reductase *nifH* gene suggested greater phylogenetic diversity among diazotrophs in methane seep sediments[36–38]. Biological nitrogen fixation requires large amounts of ATP and high-potential electrons, whereas anaerobic methane oxidation associated with ANME and SRB are among the lowest energy-yielding reactions that can sustain life[20,31,39]. From this point of view, we hypothesize that biological nitrogen fixation in cold seeps does not necessarily rely on sulfate-dependent anaerobic oxidation of methane. Considering the existence of phylogenetically and functionally diverse communities in cold seep sediments[11,40], other catabolic processes are predicted to also drive nitrogen fixation.

In this study, we investigate the hidden diversity and distributions of nitrogenases and diazotrophs, and compile evidence for their in situ activities within deep-sea cold seep sediments. To this end, gene- and genome-centric analyses of 61 metagenomes are coupled with six metatranscriptomes derived from 11 globally distributed areas of hydrocarbon seepage (Fig. 1 and Supplementary Data 1). Samples originate from five types of cold seeps, namely gas hydrates, mud volcanoes, asphalt volcanoes, oil and gas seeps and methane seeps. Most seep types have previously been shown to have lighter $\delta^{15}N$ indicative of biological nitrogen fixation, compared to nearby background sediments[13,41,42] (Supplementary Fig. 1 and Supplementary Notes). Overall, this study corroborates deep-sea cold seep sediments as overlooked habitats for uncovering diverse diazotrophs from uncultivated lineages supported by diverse energy sources, and emphasizes the importance of nitrogen fixation in a carbon-dominated environment.

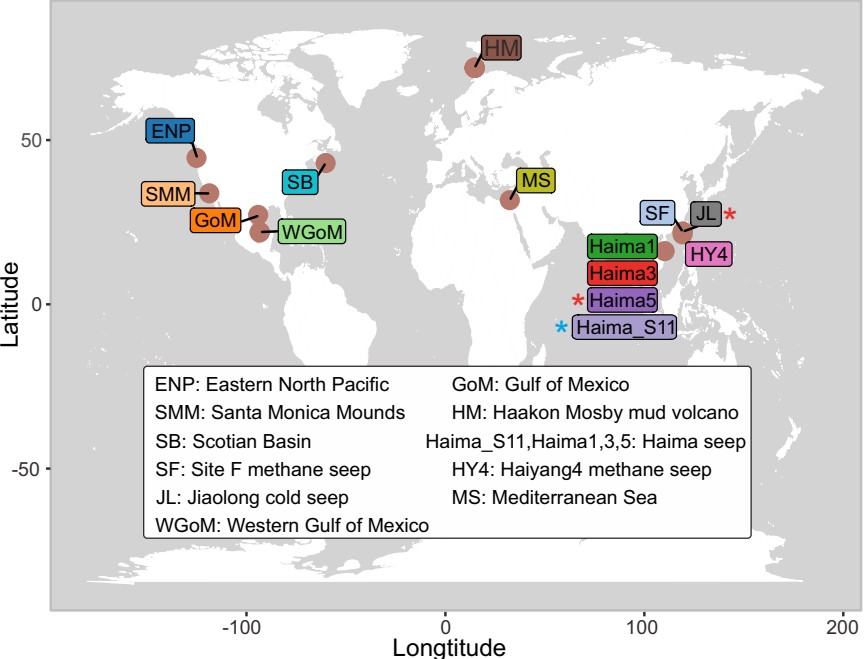

**Fig. 1 | Geographic distribution of 11 cold seep sites where metagenomic and metatranscriptomic data were collected.** These samples were originated from five types of cold seeps: gas hydrates, mud volcanoes, asphalt volcanoes, oil and gas seeps and methane seeps. Sites with red asterisks denote that both metagenomes and metatranscriptomes were collected, sites with blue asterisks denote that only metatranscriptomes were collected, and sites without asterisks denote that only metagenomes were collected. Also see details in Supplementary Data 1. The world map was drawn using the ggplot2 package in R v4.0.3.

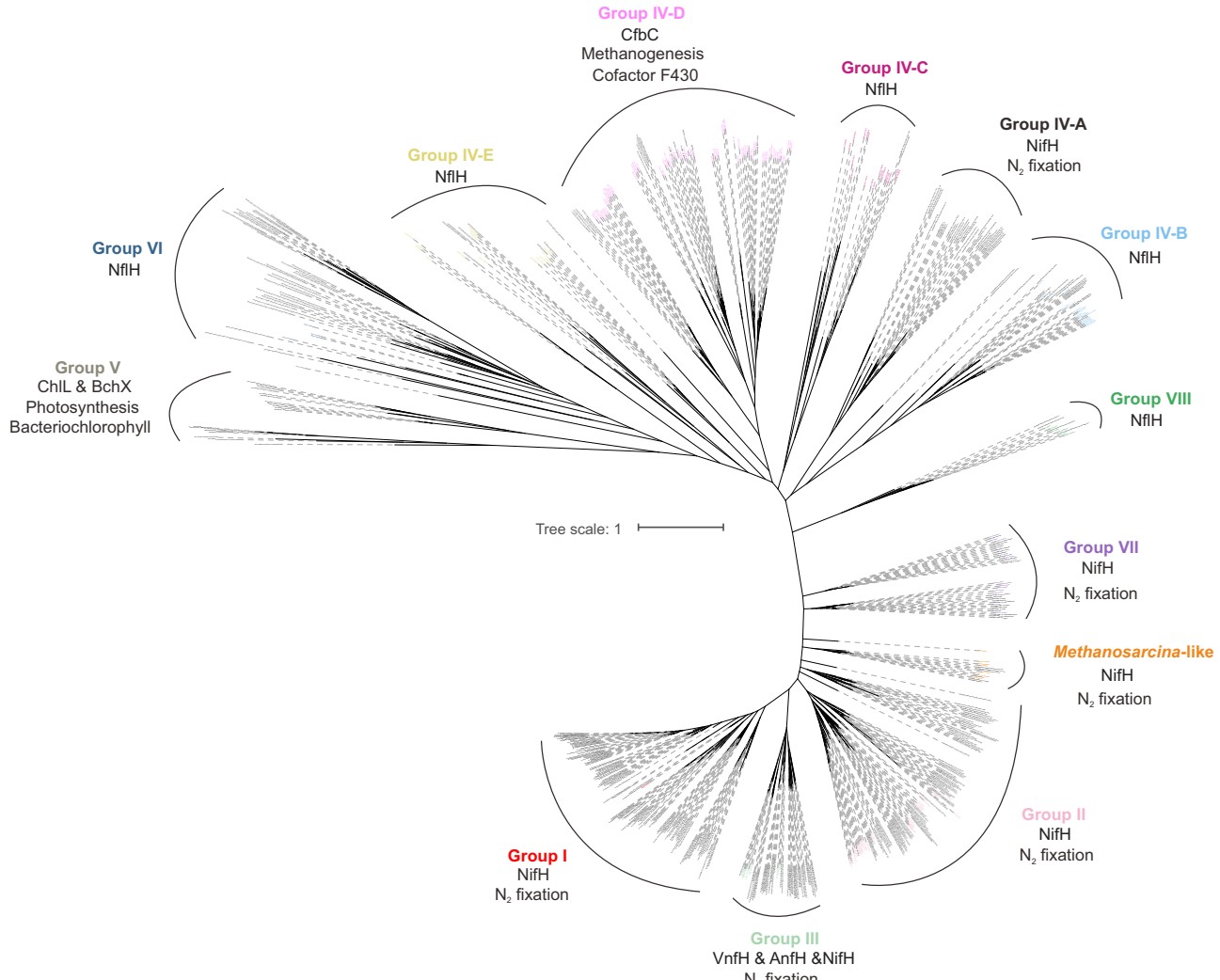

**Fig. 2 | Maximum-likelihood phylogenetic tree of non-redundant nitrogenase subunit NifH identified from cold seep metagenomic assemblies.** Homologues of *nifH* were classified into canonical groups I to III, nitrogen fixation-like groups IV to VI, and newly assigned groups including groups of *Methanosarcina*-like, VII and VIII. NflH denotes NifH-like sequences. Scale bar indicates the mean number of substitutions per site.

## Results and discussion

### Cold seeps harbor canonical and novel nitrogenase gene homologues

The *nifH* marker gene, which encodes a key structural protein of the nitrogenase enzyme, is commonly used to explore the diversity and abundance of diazotrophs in various environments[15,43]. Annotations of contigs assembled from 61 metagenomes collected at 11 globally distributed cold seep sampling stations (Fig. 1 and Supplementary Data 1) revealed 202 non-redundant *nifH* homologues falling into the nitrogenase superfamily. The phylogenetic tree (Fig. 2) suggested that *nifH* homologues were classified into distinct bona fide nitrogenase sequences (canonical groups I to III) as well as nitrogenase-like groups (groups IV to VI)[44–48]. These include (1) typical Mo-Fe nitrogenases from aerobic and facultative anaerobic bacteria (group I; $n = 1$); (2) Mo-Fe nitrogenases from anaerobic bacteria and archaea (group II; $n = 32$); (3) alternative nitrogenases (Mo-independent Anf and Vnf) and some Mo-Fe nitrogenases from Euryarchaeota[48] (group III; $n = 11$); (4) poorly characterized *nif* homologues (group IV; $n = 123$); (5) bacteriochlorophyll and chlorophyll biosynthesis genes[49] (group V; $n = 1$); and (6) putative tetrapyrrole cofactor biosynthesis genes[44] (group VI; $n = 5$). Group IV genes include its subclusters B ($n = 19$), C ($n = 19$) and E ($n = 16$) with unknown functions, as well as subcluster D ($n = 69$) involved in archaeal methionine biosynthesis[44–48]. Subcluster A within

group IV also includes functional nitrogenases found in *Endomicrobium proavitum* that can fix nitrogen[50], but none of the identified *nifH* homologues from the cold seep assemblies are affiliated with this subcluster. Despite this diversity, this classification scheme (based on Meheust et al.)[44] still did not sufficiently reflect the variety of nitrogenase genes found in cold seep diazotrophic populations. Following the approaches reported by Dekas et al.[35], Miyazaki et al.[37] and Al-Shayeb et al.[51], unclassified sequences formed three distinct lineages (Fig. 2) including (1) a clade similar to *nifH* found in *Methanosarcina* species but not clearly falling into the canonical groups (i.e., *Methanosarcina*-like group, MSL; $n = 7$), (2) a novel clade proposed here as group VII ($n = 15$), and (3) a novel clade proposed here as group VIII containing *nifH*-like genes ($n = 6$). Among the three novel lineages, MSL and group VII were considered as bona fide *nifH* based on the analyses of nitrogenase operon structure and conserved motif detailed below.

Read abundance ratios of bona fide *nifH* ($n = 66$) and single-copy ribosomal protein genes were used as a proxy for the relative abundance of putative diazotrophs in the total microbial community[52,53]. Diazotrophs were typically abundant within various types of cold seep sediments ($24 \pm 22\%$ of the total bacterial and archaeal community) (Supplementary Data 2), but this varied greatly between samples (from 0.7% at a North Pacific gas hydrate to 93% at Gulf of Mexico oil and gas

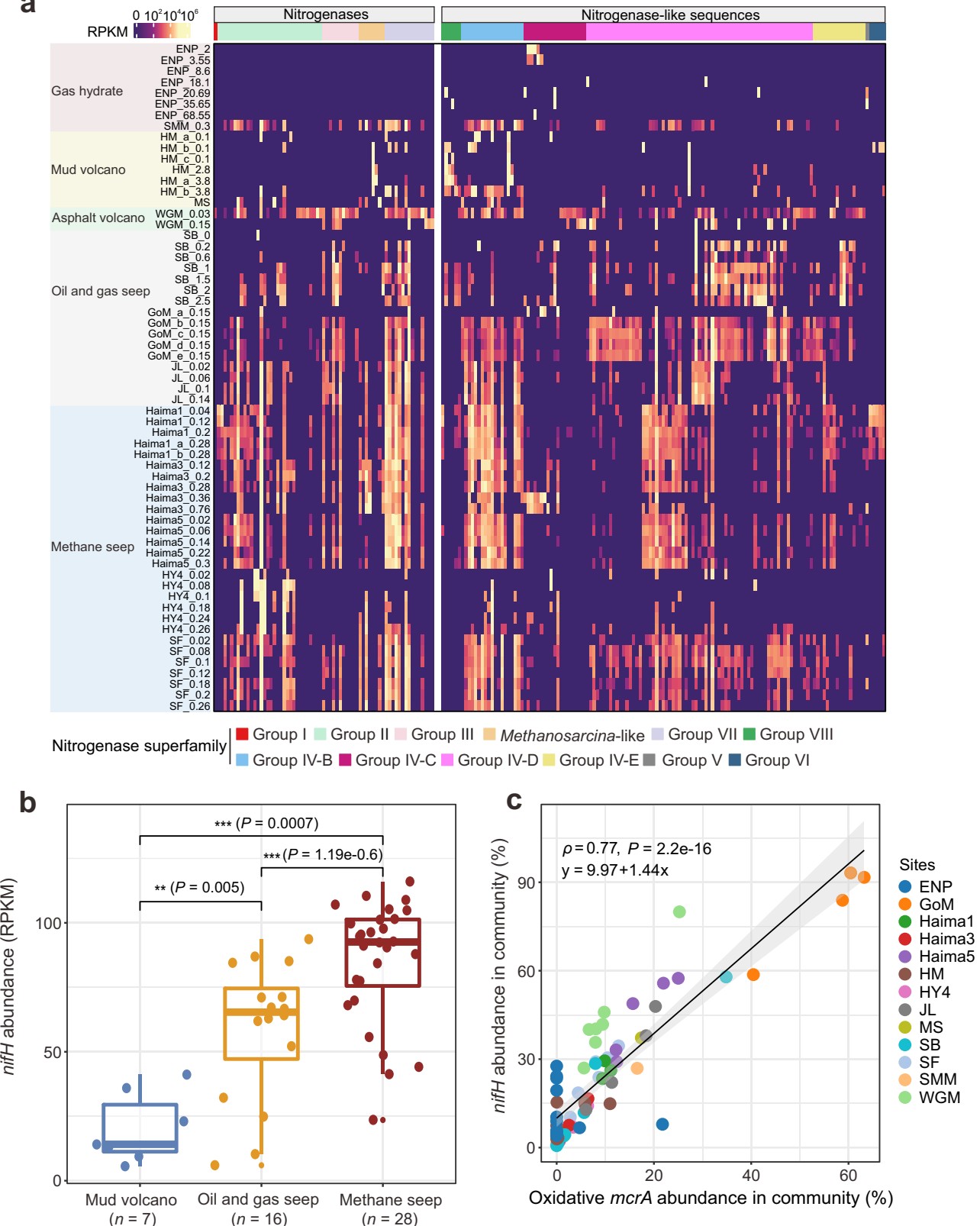

**b** ... *(P = 0.0007)*
*(P = 0.005)* ... *(P = 1.19e-0.6)*

**c** $\rho = 0.77$, $P = 2.2e-16$
$y = 9.97 + 1.44x$

samples) exhibiting correlation with seep type and sediment depth (Fig. 3a). Heterotrophic microorganisms must oxidize large amounts of organic carbon to generate sufficient ATP to fix nitrogen under anoxic conditions[43]. Accordingly, cold seeps classified as high-flux mud-prone systems, including oil, gas and methane seeps, hosted the highest densities of diazotrophs (Fig. 3a, b), suggesting a potential

control exerted by hydrocarbon flux rates on cold seep diazotrophs. Interestingly, a positive correlation was also observed between the gene abundance for *nifH* and the oxidative *mcrA* gene typical of ANME archaea (Spearman's $\rho = 0.77$, $P = 2.2e−16$; Fig. 3c). Variants of the oxidative methyl-coenzyme M reductase A (McrA) are used as indicators for estimating the relative abundance of anaerobic methane- and

**Fig. 3 | Relative abundance patterns of 202 *nifH* genes. a** Relative abundances of 202 *nifH* genes from different cold seep sediments, shown as RPKM (reads per kilobase per million mapped reads). **b** Comparison of *nifH* gene abundances in different types of cold seep ecosystems. *n* values refer to the number of biologically independent samples for statistics analysis. Asterisks indicate statistically significant differences between groups of mud volcanoes, oil and gas seeps, and methane seeps (determined by two-sided Wilcoxon Rank Sum test; * for *P* < 0.05, ** for *P* < 0.01 and *** for *P* < 0.001). Boxplot components: center line, median values;

box limits, upper and lower quartiles; whiskers, 1.5× interquartile range; points, outliers. **c** Significant Spearman correlation between relative abundances of *nifH* and the oxidative *mcrA* gene. Percentages were calculated by dividing the RPKM value of *nifH* genes by the mean of RPKM values estimated from 14 single-copy marker genes. The gray shadow indicates the 95% confidence interval. The abbreviations of the sites are shown in Fig. 1. Source data are provided as a Source Data file.

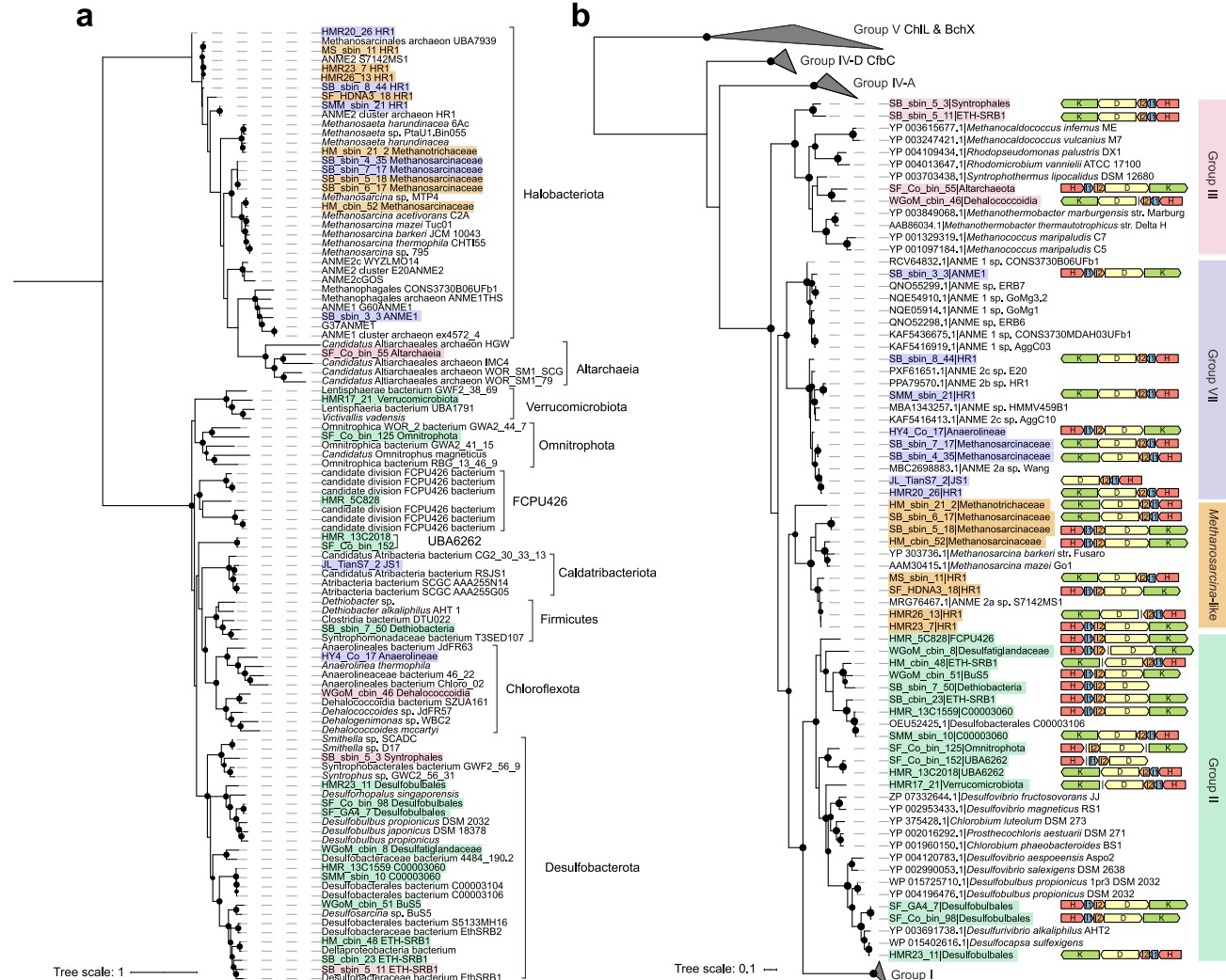

**Fig. 4 | Maximum-likelihood phylogenetic trees of nitrogen-fixing MAGs and their NifH protein sequences. a** Phylogenomic analysis of 35 MAGs containing nitrogen fixation genes. This maximum-likelihood tree is based on concatenation of 43 conserved protein sequences. MAGs are colored based on their phylogenetic affiliation at the phylum level. **b** Phylogenetic analysis of identified NifH protein sequences and genomic context of corresponding *nif* genes in the same 35 MAGs with nitrogen fixation genes. The scale bar represents one amino acid substitution per sequence position. For both trees, bootstrap values of >70% are indicated as black circles at the nodes, and scale bars indicate the mean number of substitutions per site.

multi-carbon alkane-metabolizing archaea[5,54,55]. In this context, *nifH* sequences belonging to the newly discovered clade group VII were the most abundant (Fig. 3a), highlighting the unique diversity of hydrocarbon seep diazotrophs compared to other studied ecosystems, including the deep-sea background sediments[29,56].

**Diverse diazotrophs from ten different phyla reside in cold seep sediments**

Using metagenomic assembly and binning strategies, we recovered 1428 non-redundant bacterial (*n* = 1146) and archaeal (*n* = 282) population genomes (Supplementary Data 3) belonging to 76 phyla based on the Genome Taxonomy Database (GTDB; see Methods). Most

genomes were affiliated with the phyla Chloroflexota (*n* = 239, namely Chloroflexi in NCBI taxonomy), Desulfobacterota (*n* = 185, namely Deltaproteobacteria), Halobacteriota (*n* = 114, namely Euryarchaeota), Proteobacteria (*n* = 130), Acidobacteriota (*n* = 70, namely Acidobacteria), Bacteroidota (*n* = 65, namely Bacteroidetes), Planctomycetota (*n* = 54, namely Planctomycetes), Thermoplasmatota (*n* = 48, namely Thermoplasmata), and Asgardarchaeota (*n* = 43, namely Asgard superphylum). Among these genomes, 20 bacterial and 15 archaeal MAGs spanning ten different phyla encoded nitrogenase genes (Fig. 4a and Supplementary Data 4), and belong to the Halobacteriota (*n* = 14), Desulfobacterota (*n* = 11), Chloroflexota (*n* = 2), UBA6262 (*n* = 2, candidate phylum), Altarchaeota (*n* = 1),

Caldatribacteriota ($n = 1$, namely Atribacteria), Omnitrophota ($n = 1$, namely Omnitrophica), FCPU426 ($n = 1$, candidate phylum), Verrucomicrobiota ($n = 1$, namely Verrucomicrobia), and Firmicutes ($n = 1$). Within the phylum Halobacteriota, nitrogenase-encoding MAGs span the lineages ANME-1 ($n = 1$), ANME-2 ($n = 7$), Methanotrichaceae ($n = 1$), and Methanosarcinaceae ($n = 5$). Within the Desulfobacterota, nitrogenase-encoding MAGs belonged to the order of "C00003060" (aka SEEP-SRB1c[39]) along with other non-ANME-associated bacterial groups such as BuS5 (aka *Desulfosarcina* sp. BuS5 in NCBI taxonomy), Desulfatiglandaceae and Syntrophales known to degrade alkanes or aromatic hydrocarbons coupled with sulfate reduction[7,57]. The increased diversity of bacterial and archaeal diazotrophic lineages substantially broadens the genomic database of microbial diazotrophs in deep-sea cold seep sediments[48], which previously only included ANME-2b and SEEP-SRB1g[34]. Indeed, to our knowledge, this represents the first genomic evidence of nitrogen fixation potential in five different phyla, namely Altarchaeia, Omnitrophota, Caldatribacteriota along with two bacterial candidate phyla FCPU426 and UBA6262[48,58]. Among archaea, only lineages related to anaerobic methanogens and closely related anerobic methanotrophs are known or predicted to possess nitrogenases[58,59]. Our detection of Altarchaeia and other archaeal lineages (e.g., ANME-1) as potential diazotrophs also expand the known diversity of nitrogen-fixing archaea (Fig. 4a and Supplementary Data 4).

Phylogenetic analysis of NifH (Fig. 4b) reveals that nitrogenases encoded by these 35 genomes belong to groups of II, III, MSL, and VII. A large majority of these genomes (32 out of the 35) encode *nifHDK* gene clusters for synthesis of the complete nitrogenase complex. Pairwise alignments of amino acids with bona fide nitrogenases (Supplementary Fig. 2) show that 28 identified NifH sequences contain conserved residues important for ATP hydrolysis and [4Fe4S] cluster coordination (Cys97 and Cys132)[46,60]. These NifH sequences also contain conserved residues (Arg100) for ADP-ribosylation, a reversible post-translational modification for nitrogenase activity regulation in the bona fide nitrogenases[45]. All residues required for the coordination of the P-cluster (Cys62, Cys88 and Cys154) with the Fe atom of the FeMo cofactor (Cys275 and His442) are conserved among 30 NifD sequences[45,61] (Supplementary Fig. 3). Crucial residues of the P cluster (Cys70, Cys95, and Cys153) are also conserved in 31 NifK sequences (Supplementary Fig. 4). By contrast, one or more conserved cysteine residues in the molybdenum nitrogenase subunits NifD and NifK for P-cluster coordination are absent in the bacteriochlorophyll oxidoreductase (ChlLNB and BchXYZ) and reductive cyclase of $F_{430}$ synthesis (CbfCD) systems (which both ligate a catalytic [4Fe4S] cluster instead)[46,62]. Overall, the conserved active sites observed among NifH, NifD and NifK homologues suggest that the newly assigned groups MSL and VII nitrogenases most likely function analogously to their canonical group I-III counterparts.

For these 35 MAGs, each *nif* gene cluster also contained a pair of genes downstream of *nifH* that are here designated as $nifI_1$ and $nifI_2$ (Fig. 4b). The products of $nifI_1$ and $nifI_2$ are both members from the $P_{II}$ family of nitrogen-regulatory proteins, known to switch-off nitrogenase activity at the post-translational level[63]. $NifI_1I_2$ regulatory mechanisms are typically present in anaerobes, including all diazotrophic methanogens, as well as anaerobic bacteria including *Chlorobium tepidum*, *Dehalococcoides ethenogenes* and some Desulfobacterota[64,65].

Based on read mapping, distributions of the 35 diazotrophs were compared across metagenomes obtained in different types of samples from all of the cold seeps analyzed in this study (Supplementary Fig. 5 and Supplementary Data 5). Their overall relative abundance is $4 \pm 3\%$, far below the estimated values based on read abundance ratios of *nifH* genes (Supplementary Data 2). Two possible explanations might account for this: (1) diazotrophic MAGs might contain multiple copies of *nifH*; (2) there are still some diazotrophs that we did not recover here. When considered individually, Desulfobacterota (comprising up

to 2% of the microbial community) and Caldatribacteriota (also up to 2%) represented the major bacterial diazotrophs, and Halobacteriota constituted major archaeal diazotrophs (up to 13%). While members of the Caldatribacteriota phylum are prevalent in cold seep sediments[11,40], no previous studies have inferred that they are diazotrophic. These results highlight that Caldatribacteriota may play biogeochemically and ecologically significant roles within diverse cold seeps besides their role in carbon cycling[66]. Most other diazotrophs are at lower abundance (<1% of the microbial community). Overall, it can be speculated that cold seep diazotrophs are widespread and abundant to substantially contribute to the deep-sea nitrogen balance.

## Various organic and inorganic energy sources support nitrogen fixation

With the consumption of 16 ATP molecules per dinitrogen reduced, the nitrogenase system is energetically costly for microorganisms[28]. To provide a global view of functional capabilities among the 35 diazotrophs, metabolic capabilities were annotated based on marker genes and pathways. Genomic analyses of these 35 MAGs identified four distinct groups regarding carbon cycling (Fig. 5 and Supplementary Data 6–8): (1) anaerobic methane-oxidizing archaea, including one ANME-1 and six ANME-2 (Fig. 5a); (2) hydrogenotrophic methanogens, including one Methanotrichaceae and three Methanosarcinaceae (Fig. 5c); (3) anaerobic non-methane alkane-degrading bacteria, including two Desulfobacteria (one Desulfatiglandaceae and one BuS5), one Syntrophales and one Caldatribacteria (Fig. 5d); and (4) heterotrophs capable of degrading complex organic matter, such as cellulose, chitin, glucan, pectin, polyphenols, and starch (Supplementary Data 8). With respect to electron acceptors, sulfate reduction genes were identified in eight Desulfobacterota and one Caldatribacteriota (Fig. 5b and Supplementary Data 7), in agreement with a previous report that sulfate reduction supports diazotrophy in marine sediments[29]. Metal reduction related *mtrC* was identified in the genome of ETH-SRB1 (Supplementary Data 6), suggesting that this organism may also use iron or manganese as a terminal electron acceptor[67]. Three Desulfobulbales may be capable of both nitrate reduction and nitrogen fixation based on the presence of *napA* and *napB* (Supplementary Data 6). Based on the presence of two structural genes of form I RuBisCO and various genes for the Wood Ljungdahl pathway (Supplementary Data 6), some microorganisms represented by these MAGs can function as autotrophs, suggesting potential chemolithoautotrophic diazotrophy. The genomes also exhibit the potential for further assimilation of fixed ammonium into amino acids through the sequential action of glutamine synthetase (GS) and glutamate synthase (GOGAT) enzymes or NADH-glutamate dehydrogenase (GDH)[68] (Fig. 5).

Nitrogenases not only mediate the reduction of molecular nitrogen into ammonia, but also reduce protons into molecular hydrogen during their reaction cycle[69]. Some diazotrophs identified here, including those within Caldatribacteriota, Desulfobacterota and Methanosarcinaceae (Supplementary Data 9), have the potential to internally recycle this hydrogen as an energy source, for example by using group 1 [NiFe]-hydrogenases linked to anaerobic respiratory chains[70]. Not all diazotrophs possess hydrogenases, potentially allowing non-diazotrophic bacteria to deploy uptake hydrogenases to consume hydrogen released by hydrogenase-deficient diazotrophs. The latter include various Chloroflexota, Desulfobacterota, Gammaproteobacteria, Campylobacterota, and Planctomycetota in the cold seep sediments. Thus, the nitrogenase reaction is also likely to have diverse consequences for nutrient cycling in cold seep sediments.

To infer whether the potential diazotrophs identified in the metagenomes can fix $N_2$ under in situ conditions, two metatranscriptomes sequenced from Haima cold seep sediments and four metatranscriptomes sequenced from Jiaolong cold seep sediments (Fig. 1 and Supplementary Data 1) were mapped against nitrogenase-

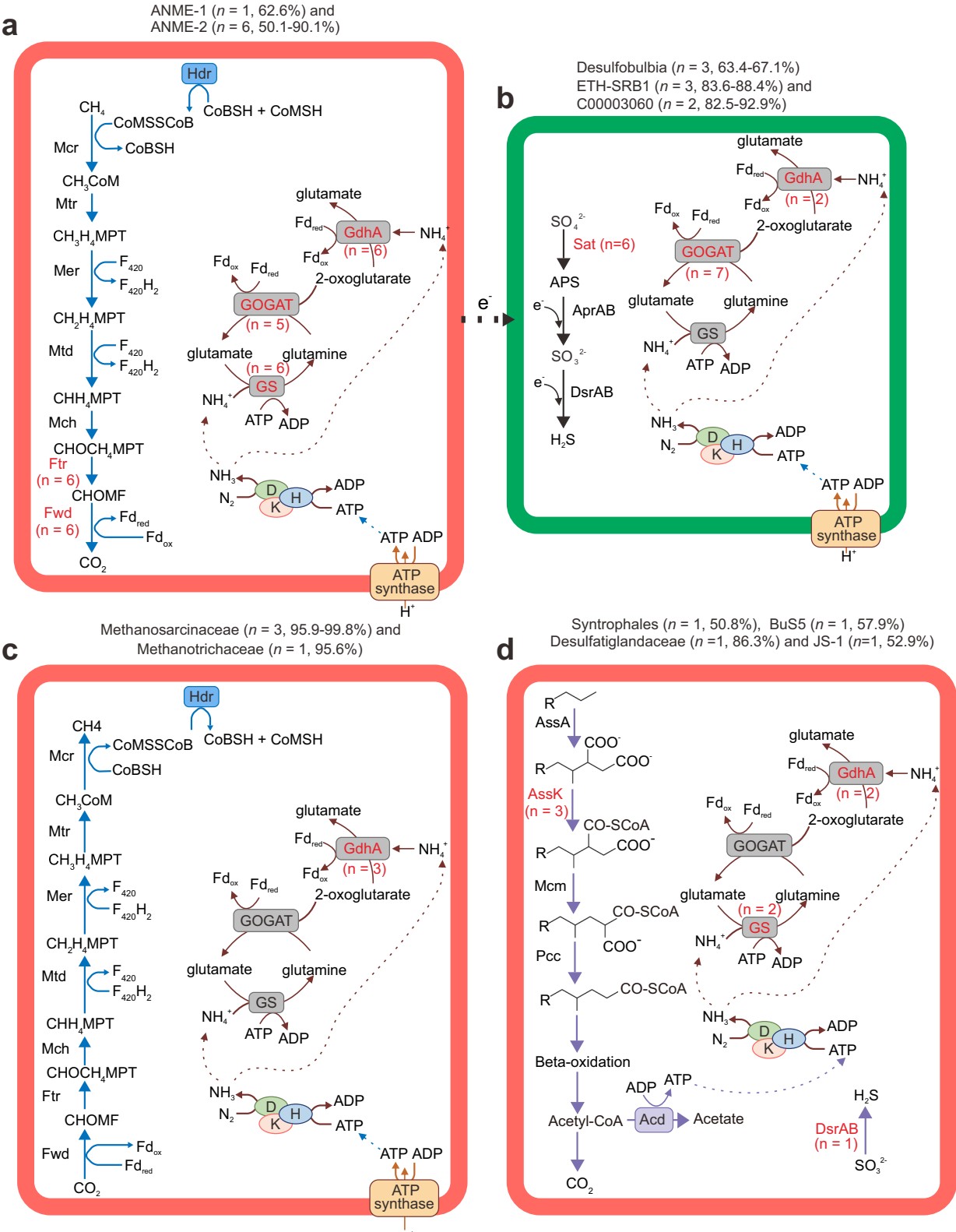

encoding MAGs. For both seep sites, the *nifH* genes of ANME-1, ETH-SRB1, and Caldatribacteriota were transcribed at moderate to high levels, up to 60–335 transcripts per million reads (TPM) (Supplementary Data 10), whereas fewer transcripts from ANME-2 were detected. Transcript levels were higher in deeper sediments relative to surficial layers, suggesting nitrogen fixation is particularly important when microbial carbon metabolism (e.g., methane oxidation) is prevalent and nitrogen oxides are limited[71]. Transcribed *nifH* genes in various microbial groups suggest that diverse catabolic processes actively fuel nitrogen fixation in cold seep sediments. This supports the hypothesis that nitrogenases have been acquired by organisms inhabiting nearly every characterized ecological niche, consistent with a selective advantage for organisms able to relieve nitrogen limitation.

**Fig. 5 | Metabolic reconstruction of core pathways for nitrogen-fixing MAGs.**
**a** Anaerobic archaeal oxidation of methane; **b** dissimilatory sulfate reduction;
**c** archaeal methanogenesis; **d** anaerobic degradation of alkanes by bacteria. Red
font indicates that not all MAGs retrieved include the gene (numbers of MAGs with
the corresponding gene indicated in parentheses). The percentages between
brackets indicate the estimated completeness of the corresponding MAGs. Mtr N[5]-
methyltetrahydromethanopterin–coenzyme M–methyltransferase, Mer 5,10-
methylenetetrahydromethanopterin reductase, Mtd methylenetetrahy-
dromethanopterin dehydrogenase, Mch methenyltetrahydromethanopterin

cyclohydrolase, Ftr formylmethanofuran-tetrahydromethanopterin N-for-
myltransferase, Fwd formylmethanofuran dehydrogenase, Hdr heterodisulfide
reductase, APS adenosine phosphosulfate, Apr adenylylsulfate reductase; Sat sul-
fate adenylyltransferase, GS glutamine synthetase, GOGAT glutamate synthase,
GDH NADH-glutamate dehydrogenase, Ass alkylsuccinate synthase, AssK CoA-
ligase, Mcm methylmalonyl-CoA mutase, Pcc propionyl-CoA carboxylase, Acd
acetate-CoA ligase (ADP-forming). Detailed enzyme annotation is presented in
Supplementary Data 7.

## *nif* genes are subject to mobile genetic element transfers and purifying selection

Microorganisms can acquire genes through horizontal gene transfer (HGT), which enables them to adapt to changing environmental conditions and thus occupy expanded ecological niches[72]. Previous studies have suggested that HGT events have crucially impacted the distribution of nitrogenase genes[45,48,73]. For example, thermophilic Aquificales acquired the ability to fix N$_2$ from thermophilic Deferribacteres[56], and the acquisition of Nif by Firmicutes possibly arose through a HGT event with an ancestral methanogen[58,74]. Gene neighborhood analyses of MAGs from cold seeps revealed gene clusters of mobile genetic elements (MGEs) together with *nifHDK*, nitrogenase regulation and metal cofactor biosynthesis genes, and molybdenum/molybdate and ammonium transporter genes (Fig. 6). The MGEs identified here included retrotransposable and transposable elements, with the former transferred via an RNA intermediate[75] between host genomes and the latter serving as mobile DNAs[72]. Five diazotrophic MAGs contained genes for retrotransposable elements, including reverse transcriptase, endonuclease, DEAD/DEAH box helicase and nucleotidyltransferase. A transposon gene, integron integrase, was only found in one ETH-SRB1 genome. MGEs near *nif* gene clusters have been previously reported in other diazotrophs, indicating HGT[26,60]. To integrate into the host genome, MGEs require ATP hydrolysis[76]. Accordingly, AAA-type ATPase genes are interspersed with retrotransposable elements in the *nif* gene cluster of UBA6262 HMR_13C2018. These proteins have been biochemically demonstrated to control efficient transposition through DNA remodeling and transposase recruitment[77]. Meanwhile, the phylogenies of most NifH sequences were observed to be inconsistent with their corresponding taxonomies (Fig. 4). Except for *Methanosarcina*-like group NifH, sequences from group II, group III and group VII were scattered among diverse bacterial and archaeal phyla (Fig. 4). For example, six different bacterial phyla encoded NifH sequences of group II. Combined with the MGEs analysis, these results suggest that HGTs occurred among cold seep communities during their evolution. Nevertheless, vertical transmission of these genes in deep-sea cold seeps, like what has been observed for nitrogenases in the surface ocean, cannot be ruled out[25].

Intra-population genetic diversity (i.e., microdiversity) may increase the fitness of a genotype in ecosystems with changing conditions. InStrain[78] was used to assess within-sample *nif* microdiversity based on metagenomic paired reads. Genomic nucleotide diversity (π) was calculated based on all reads, and as the average number of nucleotide differences per base pair for *nifHDK* genes. The observed nucleotide diversity was low, ranging from zero to 0.04, and mostly varied without significance between five different cold seep types (Fig. 7a). This indicates that these genes are highly conserved both across and within samples, regardless of sampling location, possibly because few mutations accumulated during sediment burial[79]. Nucleotide diversity was also estimated to be similar among *nifH*, *nifD* and *nifK* genes (Fig. 7a). The ratios of the two rates of non-synonymous to synonymous polymorphism (pN/pS) in *nifHDK* were determined (Fig. 7b) to assess if genes are under purifying (negative) selection which involves the selective removal of deleterious mutations[79]. In general, pN/pS ratios below 1 indicate that a gene is under selective pressure to remove deleterious mutations to maintain protein

function[80]. Calculation values were all well below 1 (between 0.02 and 0.5), suggesting that *nifH*, *nifD* and *nifK* genes are under strong purifying selection in cold seep sediments[81]. This is consistent with previous studies, as generally microbial genes encoding key functions will undergo higher purifying selection compared to genes that are dispensable[80].

## Conclusion

In the deep-sea cold seep sediments that are impacted by darkness, low temperatures, and high hydrostatic pressure, growth of microbiomes consuming rich hydrocarbons is also supposed to be nitrogen limited. Biological nitrogen fixation is one main source of bioavailable nitrogen, offsetting localized nitrogen limitation and promoting ecosystem productivity. The present work demonstrates the diversity, abundance, and distribution of diazotrophs at cold seeps, revealing this metabolic guild to be diverse, widespread and probably sufficiently abundant to influence deep-sea benthic nitrogen cycling. To our knowledge, most diazotrophs detected in these cold seeps belong to candidate phyla, including the first known diazotrophs with the Altarchaeia, Omnitrophota, FCPU426, Caldatribacteriota and UBA6262. Of the 35 recovered diazotrophic MAGs, 23 represent microorganisms that are involved directly or indirectly in hydrocarbon metabolisms, including anaerobic methane-oxidizing archaea and anaerobic non-methane alkane-degrading bacteria. The tight correlation between hydrocarbon-derived carbon and nitrogen cycles indicates that nitrogen fixation pathways might be selected for microorganisms making use of the most abundant energy source at cold seeps. Moreover, we show that HGTs and purifying selection mediate cold seep nitrogenase evolution. Overall, the findings in this study highlight the importance of exploring the diversity and activity of diazotrophs in deep-sea benthic ecosystems and suggest cultivation of novel diazotrophs from cold seep sediments should be possible.

## Methods

### Compilation of δ[15]N records of bulk sediment organic matter from published literature

The previously published δ[15]N records of bulk sediment organic matter from five active cold seep sites were compiled (Supplementary Data 11), including the Napoli and Amsterdam mud volcanoes in the eastern Mediterranean Sea, an oil and gas seep in the Northern Gulf of Mexico, and methane seeps in the South China Sea (Site F and Haima; both in the vicinity of gas hydrate deposits). To eliminate the influence of sources of organic matter, the δ[15]N values of background sediments nearby these seep sites were also compiled and used as controls.

### Metagenomic datasets for deep-sea cold seep sediments

Metagenomes were compiled from 61 deep-sea sediment samples (water depths ranging from 860–3005 m) collected from 11 geographically diverse cold seep sites from around the world (Fig. 1, Supplementary Data 1, and references therein). Part of these metagenomes were downloaded from NCBI's Sequence Read Archive, including datasets derived from Haakon Mosby mud volcano, Eastern North Pacific ODP site 1244, Mediterranean Sea Amon mud volcano, Santa Monica Mounds, and Gulf of Mexico[66,82–85]. Other data were obtained from our previous publications described in detail

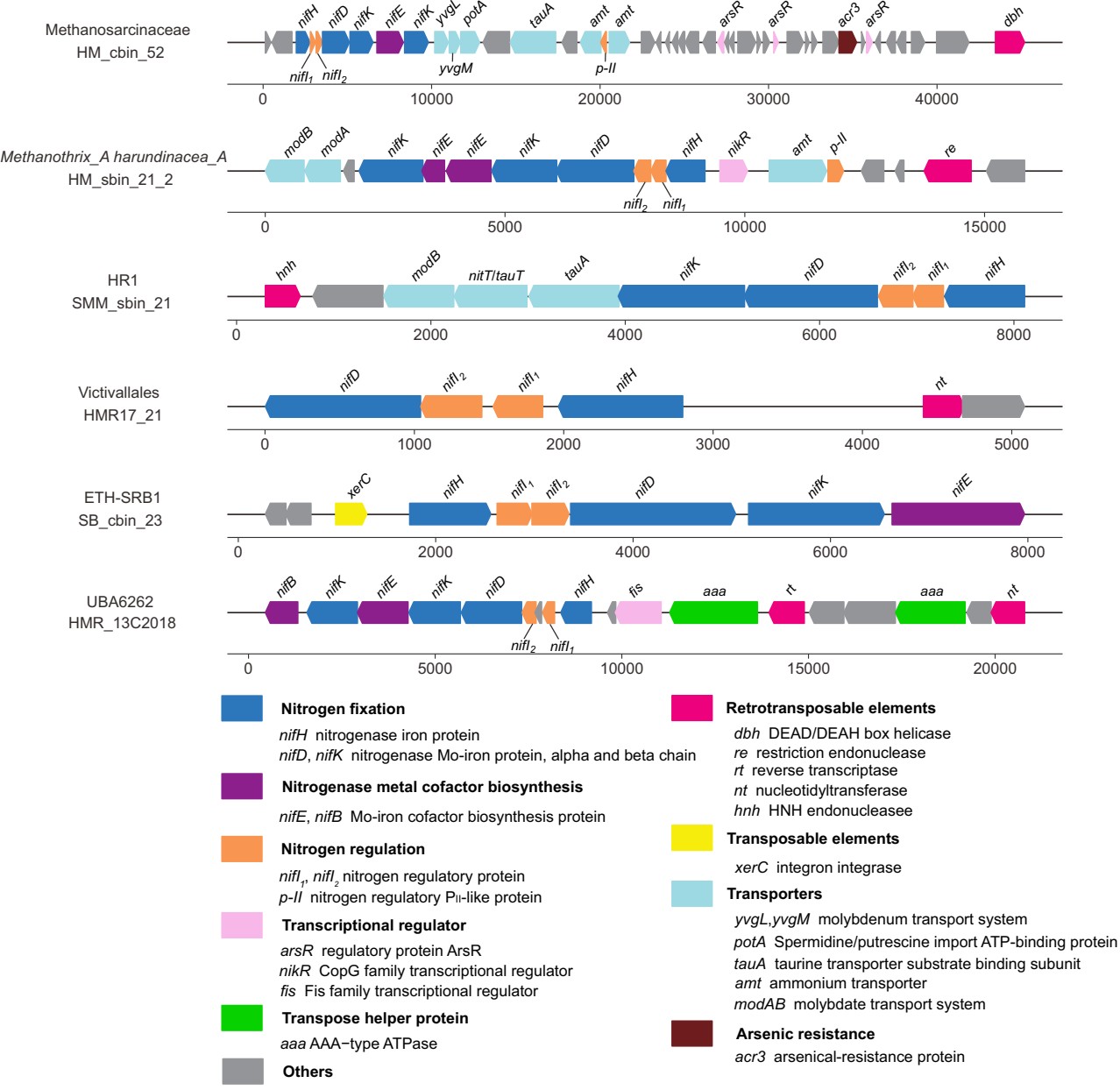

**Fig. 6 | Genomic context of nitrogen fixation genes.** Gene neighborhoods of *nifHDK* include retrotransposable or transposon elements, regulatory nitrogen fixation genes, nitrogenase metal cofactor biosynthesis genes and transporter genes. Source data are provided as a Source Data file.

elsewhere[11,40,71,86,87], including a Scotian Basin cold seep in the northwest Atlantic Ocean, and the South China Sea cold seeps Jiaolong, Haiyang4, Site F and Haima (Supplementary Data 1).

**Metagenome assembly and binning**
Metagenomic raw reads were checked for quality, assembled and binned using the metaWRAP v1.3.2 pipeline[88]. In brief, raw reads were trimmed using the metaWRAP Read_QC module (parameters: -skip-bmtagger). For each cold seep site, quality-controlled reads were individually assembled and co-assembled using the metaWRAP Assembly module (parameters: -megahit)[89]. Short contigs (<1000 bp) were removed. Each metagenomic assembly was binned using the metaWRAP Binning module (parameters: -maxbin2 -concoct -metabat2). The three bin sets for each assembly were consolidated using the metaWRAP Bin_refinement module (parameters: -c 50 -x 10)[88]. As an exception, filtered reads from the Haima cold seep site were individually assembled and binned using the same method mentioned

above. All individual assemblies from Haima site were then concatenated and binned using the VAMB tool (v3.0.1; default parameters)[90]. All produced bins were aggregated and dereplicated to a non-redundant set of strain-level metagenome-assembled genomes (MAGs) using dRep (v2.6.2; parameters: -comp 50 -con 10)[91] at 99% average nucleotide identities[92]. Completeness and contamination of MAGs were evaluated using CheckM v1.0.18[93]. Additionally, we used GUNC v1.0.1[94] to assess chimerism and contamination of the diazotrophic MAGs.

**Taxonomic classification of MAGs**
Taxonomy assignment of each MAG was initially performed using GTDB-TK v1.5.1[95] with reference to GTDB R06-RS202 and then validated using a maximum-likelihood phylogenomic tree. Reference genomes accessed from NCBI GenBank and the MAGs from this study were used to construct the phylogenomic tree based on concatenation of 43 conserved single-copy genes extracted by CheckM

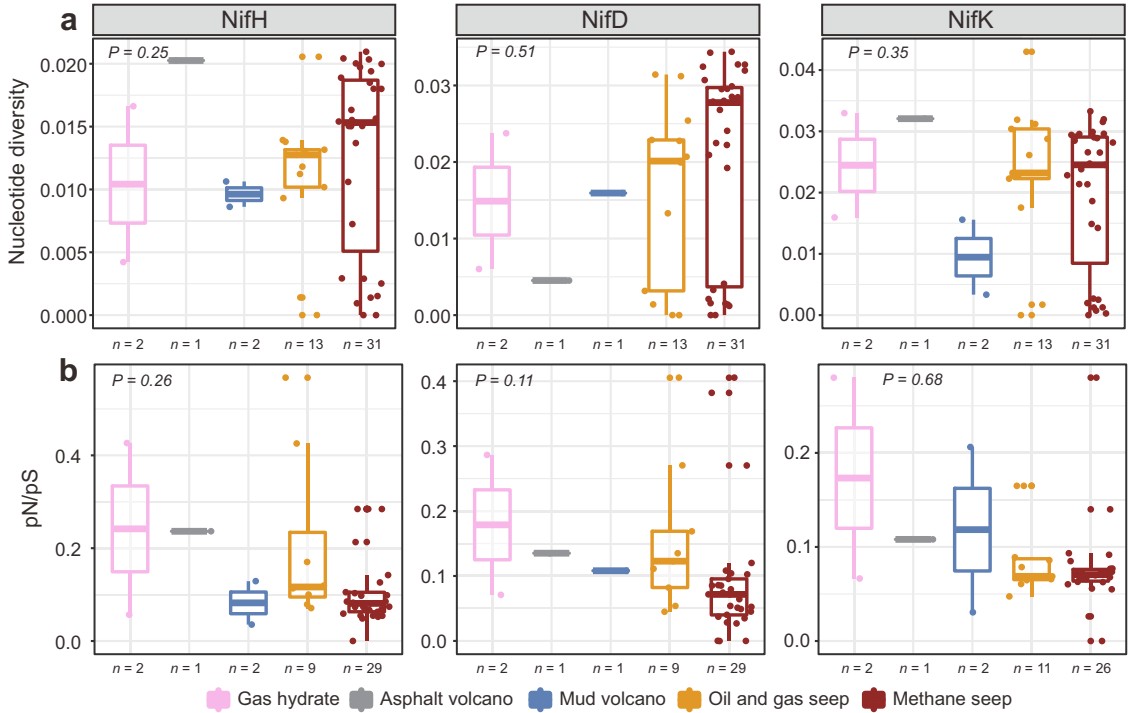

**Fig. 7 | Evolutionary metrics of nitrogen fixation genes. a** Nucleotide diversity (π) of *nifHDK* genes at different types of cold seeps; **b** pN/pS ratio of *nifHDK* genes at different types of cold seeps. Nucleotide diversity is used to measure genetic diversity within a population (microdiversity), which is calculated using the formula: $1 - [(\text{frequency of A})^2 + (\text{frequency of C})^2 + (\text{frequency of G})^2 + (\text{frequency of T})^2]$. pN/pS is the ratio of non-synonymous to synonymous polymorphism rates within a population. *n* values refer to the number of biologically independent samples for statistics analysis. The significances were analyzed by two-sided Kruskal–Wallis Rank Sum test. Boxplot components: center line, median values; box limits, upper and lower quartiles; whiskers, 1.5× interquartile range; points, outliers. Source data are provided as a Source Data file.

v1.0.18[93], following procedures described previously[40]. The maximum-likelihood phylogenomic tree was built using RAxML v8[96] with the PROTCATLG model, bootstrapped with 1000 replicates. Genomes were classified using the naming system of the GTDB taxonomy.

**Functional annotation**
For functional profiling of unassembled metagenomes, quality-controlled reads were searched against custom protein databases of representative NifH and McrA sequences (https://doi.org/10.26180/c.5230745) using DIAMOND v0.9.14 (percentage identity >50%, query coverage >80%)[97]. For this analysis, only reads with lengths over 80% of average length (bp) were used. Read counts for NifH and McrA were converted to reads per kilobase per million (RPKM) to normalize data taking into account gene length and metagenome size. The same metagenomic reads were also screened for 14 universal single-copy ribosomal marker genes used in GraftM v0.12.2[53] by DIAMOND (query coverage >80%, bitscore >40)[97] and normalized as above. Subsequently, dividing the RPKM of each gene by the average RPKM of 14 universal single-copy ribosomal marker genes provided the estimated percentage of the community with the gene, assuming one copy per genome.

For metagenomic assemblies, functional annotation was undertaken with MetaErg v1.0[98] against Pfam and TIGRFAM databases. Genes annotated as *nifH* in each assembly were compiled and clustered at 95% sequence identity using CD-HIT v4.8.1[99] to remove gene redundancy. A phylogenetic tree was constructed (see below for details) to validate findings and determine the phylogenetic clades of NifH proteins. This resulted in a set of 202 non-redundant unique *nifH* and *nifH*-like genes.

For individual MAGs, gene predictions and metabolic process analyses were performed using MetaErg v1.0[98]. Annotations were also curated against the KEGG GENEs database using GhostKOALA v2.2[100] (genus_prokaryotes + family_eukaryotes), METABOLIC v4.0[101], and DRAM v1.0[102]. Genes involved in anaerobic hydrocarbon degradation were screened using BLASTp (identity >30%, coverage >90%, $e < 1 \times 10^{-20}$) against local protein databases[11,40].

**Phylogenetic analyses of individual protein sequences**
Each individual tree was built as follows. Amino acid sequences were aligned using MAFFT v7.471 (-auto option)[103]. Alignments were further trimmed using TrimAl v1.2.59 (-gappyout option)[104]. Maximum-likelihood trees were constructed using IQ-TREE v2.0.5[105], with best-fit models and 1000 ultrafast bootstraps. The produced trees were visualized and beautified in Interactive tree of life (iTOL; v6)[106].

**Conserved residues and motifs**
Pairwise alignment of NifH, NifD, and NifK superfamily sequences for conserved active site residue analysis was performed using MAFFT (EMBL-EBI)[107] and visualized with Jalview[108].

**Abundance profiles**
RPKM values were used to represent relative abundances of *nifH* and *nifH*-like genes. The RPKM value of each *nifH* and *nifH*-like gene was calculated using CoverM v0.4.0 "contig" (https://github.com/wwood/CoverM) (parameters: -min-read-percent-identity 0.95 -min-read-aligned-percent 0.75 -trim-min 0.10 -trim-max 0.90 -m rpkm). For genome-centric analyses, *nif*-containing genomes were further dereplicated at species level (i.e., 95% ANI) to avoid arbitrary mapping between representatives of highly similar genomes. The relative abundance of each MAG was obtained using CoverM v0.4.0 "genome" (parameters: -min-read-percent-identity 0.95 -min-read-aligned-percent 0.75 -trim-min 0.10 -trim-max 0.90 -m relative_abundance).

## Expression of *nifH* genes at Haima and Jiaolong sediments

One sediment layer from the Haima cold seep (Haima5_0.3, 28–30 cmbsf) was used for metatranscriptomic extraction and sequencing (Fig. 1 and Supplementary Data 1). Total RNA was extracted from ~2.5 g of sediments using the RNeasy PowerSoil Total RNA Kit (Qiagen) according to the manufacturer's instructions. Total RNA extracts were treated with DNase I (Vazyme, Nanjing, China) to remove DNA. RNA concentrations were evaluated on Qubit 2.0 Fluorometer (Invitrogen, Carlsbad, CA, USA). The quality of RNA was checked using gel electrophoresis. RNA was reverse transcribed to cDNA using Ovation Ultralow System V2 kit (NuGen, San Carlos, CA, USA). Metatranscriptomic libraries were prepared with the VAHTS Universal V8 RNA-seq Library Prep Kit (Vazyme, Nanjing, China) and sequenced on an Illumina NovaSeq 6000 platform. Four previously sequenced metatranscriptomes from Jiaolong (JL) cold seep sediments[71] (JL_0.02, 0–2 cmbsf; JL_0.06, 4–6 cmbsf; JL_0.1, 8–10 cmbsf; JL_0.14, 12–14 cmbsf) and one metatranscriptome from the Haima cold seep (S11_6, 160–170 cmbsf) in our recent study[109] (Fig. 1 and Supplementary Data 1) were also included in the metatranscriptomic analysis. All raw metatranscriptomic reads were quality controlled using the metaWRAP Read_QC module within metaWRAP v1.3.2 pipeline[88]. SortMeRNA v.4.3.4[110] was used to remove rRNA reads. Metagenomic reads were mapped against *nifH*-containing MAGs, with those having >5× coverage at Haima and JL cold seep sites retained for further transcript per million (TPM) calculations. The TPM of metatranscriptomic clean reads mapped to predicted genes from *nifH*-containing MAGs were calculated using Salmon v0.13.1 (-meta -validateMappings)[111].

## Detection of mobile genetic elements

Contigs containing the *nifHDK* cluster were annotated against the NCBI non-redundant (NR) protein database using BLASTp (identity >60%, coverage >90%, e < 1 × 10$^{-5}$). Mobile genetic elements located on *nifHDK*-containing contigs were identified according to the results of NR annotation.

## pN/pS ratio and nucleotide diversity analyses

For producing BAM files, all metagenomic filtered reads were mapped to an indexed database of the *nif*-containing genomes using Bowtie 2 (v2.2.5; default parameters)[112]. Mapping files were then taken as input by inStrain (v1.3.1; default parameters)[78] "profile" to calculate the nucleotide diversity and pN/pS ratio at the gene level. To do the gene-level profiling, genes were called by the program Prodigal (v2.6.3; -p meta)[113] for each MAG.

## Statistical analyses

All statistical analyses were carried out in R v4.0.3. Normality of data was evaluated using Shapiro–Wilk tests before statistical analysis. For comparison of cold seep and background groups, δ$^{15}$N values of sedimentary organic matter were tested using Student's *t* test. For comparison of different types of cold seeps, *nifH* abundance was tested using Wilcoxon Rank Sum test; nucleotide diversity and pN/pS ratio were tested using Kruskal–Wallis Rank Sum test. Spearman correlation was performed using ggpmisc package v0.3.6 to assess the relationship between the abundance of *nifH* and oxidative *mcrA* genes.

## Data availability

Assemblies, reference gene catalog, MAGs, files for the phylogenetic trees have been uploaded to figshare (https://figshare.com/s/aaf49b2441cdcb26027a). The raw sequencing reads generated in this study have been deposited in NCBI under BioProject ID PRJNA831433. All other data supporting the findings of this study are available within the article and its Supplementary Information Files. The databases used in this study include GTDB database R06-RS202 (https://data.gtdb.ecogenomic.org/releases/release202/), Pfam (https://pfam.xfam.org/), TIGRfam (https://tigrfams.jcvi.org/cgi-bin/index.cgi), KEGG GENES database (https://www.genome.jp/kegg/genes.html), NCBI non-redundant protein database (https://ftp.ncbi.nih.gov/blast/db/), and the custom protein databases of representative NifH and McrA sequences (https://doi.org/10.26180/c.5230745). Source data are provided with this paper.

## Code availability

The present study did not generate codes, and mentioned tools used for the data analysis were applied with default parameters unless specified otherwise.

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

## Acknowledgements

The work was supported by the National Natural Science Foundation of China (No. 41906076 to X.D. and No. 41476104 to Y.W.), the Scientific Research Foundation of Third Institute of Oceanography, MNR (No. 2022025 to X.D.) and the Science and Technology Projects in Guangzhou (No. 202102020970 to X.D.). The study would not have been possible without publicly available genomic data from already published studies, and the authors wish to acknowledge all the participants, organizations, and funding agencies that contributed to the collection and analysis of all the data utilized in our study, including those acknowledged in refs. 66,82–85. Detailed references are listed in Supplementary Data 1.

## Author contributions

X.D. designed this study. X.D., C.Z., Y.P., and C.G. analyzed omic data. X.D., C.Z., Y.P., L.D.S., and G.W. interpreted data. H.X.Z. performed transcriptomic experiments. Y.W. and C.R.J.H. contributed to data collection. X.D., C.Z., C.R.J.H., and C.G. wrote the paper, with input from other authors.

## Competing interests

The authors declare no competing interest.
