## [Peer Review File · Nature Communications]

Reviewer comments, first round -

Reviewer #1 (Remarks to the Author):

This study addresses an often overlooked metabolic feature of carbon-dominated environments like deep-sea hydrocarbon (oil and gas) seeps: the source of nitrogen for the seep associated microbial communities. Here, the authors analyzed the presence of nitrogen fixation genes in metagenomes from several cold seeps. A relatively high diversity of *nif* genes was found, including some novel clades. N₂ fixation appears to occur frequently in microorganisms involved in hydrocarbon oxidation, establishing a link between the C and N cycles at seeps sites.

This is an actual, much needed work. It reveals the diversity of N₂ fixers in deep sea habitats, the phylogenetic identity and energy metabolism of the harboring microorganisms. A criticism is that it is based purely on metagenomics analyses; transcriptomics, which is a logical step to demonstrate expression of potential function, was done for some of the sites, but not for all. In this respect, I believe some of the claims regarding novel nitrogenase genes (e.g. L 133) should be somewhat toned down (more in the specific comments): it is not clear if these are part of functional nitrogenases. Maybe more supporting arguments can be provided by the genetic context (Figure 4b), but the authors should bring this forward.

A second major issue is the first chapter of the Results and Discussion and Figure 1: this is not original work, it is a compilation of former publications, with additional statistical analyses – in a sense what one could expect to find in a review manuscript. I understand that in the absence of direct measurements this is meant to show that N₂ fixation is generally an important process at seep sites – this is fine; however, I'm not convinced the R & D section should start with a literature overview: it is in part misleading, since it leaves the impression of original data (for example, much of the Figure 1 legend does not hint that this is a literature summary). In my opinion, this information (including citations of the original papers) belongs to the Introduction as a background and motivation for the current work.

A second argument for the above is that Figure 1 shows ¹⁵N isotope data of sites which were not part of the current metagenome work, like the Napoli and Amsterdam mud volcanoes (if one compares the Supplementary Tables 1 and 2). Including it as part of the present study leaves the impression of disconnected parts: they can hardly be directly correlated to the metagenome work, since they come from other sites or other sampling events.

L 43-46: I suggest to split this sentence for clarity

L 52-54: One of the initial studies showing alkane oxidation at seep sites is Kleindienst et al., ISME J 2014, 8:2029-2044

L 91: I believe that 'high-energy electrons' has another meaning than the context here. Perhaps high-potential electrons?

L 102-105: citations are needed here; the first section of R & D can be integrated here

L 110: what is 'purifying selection'?

L 133: I don't think phylogenetic clustering alone can be used to give a verdict if these genes encode nitrogenases or not. Many of the genes retrieved here are *nifH* homologues (according to L 150)

L 143: facultative (?) bacteria?; same line, delete 'anaerobic' in front of 'Mo-Fe nitrogenases'.

L 158-161: strictly speaking, nitrogenases are enzyme complexes, composed of several subunits, *NifH* being one of them. According to Figure 2, these are *NifH* or *NifH*-homologues clusters, not nitrogenase clusters.

L 176-178: the first paper showing Mcr involvement in multi-carbon alkane oxidation is Laso-Perez et al., Nature 2016, 539:396-401

L 191 and elsewhere in the manuscript refers to Halobacteriota, which is shown in Figure 4a to include *Methanosarcina* and ANME; according to NCBI Taxonomy, *Methanosarcina* are part of Methanomicrobia, not Halobacteriota. Am I maybe missing something some recent revision of taxonomy?

L 201: genomic evidence of nitrogen fixation: since this is based on gene presence only, it should be slightly toned down. In the absence of process demonstration, presence of genes indicates potential.

L 208: replace *nifH* (gene) with *NifH* (protein), in accordance with Fig 4b

L 238 and forward: I am not familiar with the use of the term 'recruitment' in this context. Does it refer to relative abundance?

L 258 and elsewhere: should refer to BuS5 as strain BuS5 or *Desulfosarcina* sp. strain BuS5

Section starting on L 275: without reference to works demonstrating that H₂ production during N₂ fixation can be quantitatively relevant in as much as to have an impact on the overall metabolism of other microorganisms, I find this whole section rather speculative; it is not supported by data obtained in this study: the cited table (Supplementary Table 10) is a list of H₂ases, does not show quantitative production of H₂ during N₂ fixation.

L 339 and forward: still not clear what 'purifying selection' means

The Conclusion section brings little new, overarching interpretation of the data, and it reads more like a rehearsal of previous statements. The sentence at L 353, 'These diazotrophs regularly transcribe *nifH* ...' is not really supported by the data presented here. I am wondering if the authors could comment on the apparent correlation between the hydrocarbon-derived C and N cycles, since 23 out of the 35 diazotrophic MAGs recovered (if I was correctly reading Figure 5) are of microorganisms involved directly or indirectly in hydrocarbon oxidation. Could one speculate that selective pressure was 'pushing' N₂ fixation genes into the microorganisms making use of the most abundant energy source at seeps?

Figure 4: it would be helpful for the reader if panels a and b are correlated by using similar colors for the same groups.

Figure 6: maybe I missed that, but it seems this figure is not cited in the text

Reviewer #2 (Remarks to the Author):

In the manuscript entitled "Novel diazotrophs enhance productivity of cold seep sediment ecosystems in the deep sea", Xiyang Dong, Chuwen Zhang and colleagues analyzed 61 metagenomes and fewer metatranscriptomes from 13 globally distributed deep-sea cold seeps in order to expand our understanding of nitrogen fixation by various diazotrophic taxa in these environments. The authors assessed the phylogeny of *nifH* genes from their metagenomic assemblies (gene-centric) and subsequently characterized a considerable number of environmental genomes (genome-resolved metagenomics), 35 of which contain nitrogenase genes. Authors demonstrated with genomics that a series of phyla are capable of performing nitrogen fixation, an important contribution to the field. They further explored the metabolic capabilities of those diazotrophs to understand how they acquire energy needed for nitrogen fixation. I particularly appreciated information presented in the figure 4 and more broadly the considerable efforts made by the authors to convey their insightful results throughout the manuscript. Finally, I note that key data produced in this study (includes the assemblies and MAGs) have been made public.

I did not identify any potential flaw. I have two comments (the second is optional) that might be considered to improve the strength of the study, followed by a series of smaller points.

Comment one: The manuscript is well written, and the introduction clearly describes our current understanding of nitrogen fixation in cold seeps. However, the last paragraph of the introduction is a mix of introduction (first sentence), results (description of the samples) and discussion (summary of results). In my view, authors should remove results and discussion from this section, and simply describe briefly that they studied publically available metagenomes and metatranscriptomes from various seeps to better understand taxa contributing to nitrogen fixation in these ecosystems using gene and genome-centric analyses coupled with $\delta^{15}\text{N}$ records of bulk sediment organic matter.

Comment two: Authors performed automatic binning on metagenomic assemblies and co-assemblies without any visualization and curation of especially the diazotroph MAGs. As an extra-step used in similar studies but not here, the 35 diazotroph MAGs could be imported into a platform with enhanced visualization capabilities dedicated to genome-resolved metagenomics for curation purposes, thus enhancing confidence in the quality of MAGs. While optional given the current status of our standards, providing a database of 35 manually curated diazotrophic MAGs for cold seeps could have been viewed as a better reference in years to come. At times, errors of automatic binning can be surprising...

Smaller points:

Ln 70-74: References regarding the nitrogen fixation by Cyanobacteria and heterotrophic bacteria in the Surface Ocean and deeper layers of the oceans (introduction) might be improved to better reflect the literature. Authors should make sure references for the deeper layers only cover studies for the deeper layers (I could see at least one problem there).

Ln 162: Number of nifH genes considered bona fide is required here to better understand the methodology.

Ln 165: Should be "total bacterial community", or "total bacterial and archaeal community" depending on the single copy core genes used. This is assuming that eukaryotes and viruses are excluded from the analysis.

Ln 185: It is ok to use newly defined phylum named by GTDB but it should be stated that it is GTDB taxonomy and not the commonly used NCBI taxonomy (e.g., Planctomycetota for planctomycetes).

Ln 189: 35 out of >1300 MAGs contain nitrogenase genes. Are those the most abundant ones, or is that in clear contrast with the estimate of diazotroph proportion within the community as assessed from nifH genes and single copy core genes? Needed would be the proportion of mapped reads for the 35 diazotrophic MAGs compared to all MAGs across metagenomes. If the proportion is far below 20% then there is incoherence that should be discussed. However, I agree with the authors that both the nifH and MAG analyses emphasise that some diazotrophs are very abundant in those systems, hence the relevance of such study.

Ln 310: Figure 6 is not referred to in the main text. It could be cited here.

Ln 333: it is unclear what the authors refer to with accumulation of mutations.

Ln 356: impact of HGTs is not demonstrated. Authors should modify their statement. The genes simply co-occur in the same genomic regions. If authors want to claim HGT, they need to show incoherence between the phylogenomics of genomes and the phylogeny of nifH genes. As far as I could see from the figure 4, the coherence between the two is quite important. I would like a further explanation by the authors on this matter.

Tom O. Delmont

Reviewer #3 (Remarks to the Author):

This is a meta-analysis of published metagenomes for diazotrophy-relevant genes, supplemented with some geochemical information (N stable isotopes) on the sampling sites. Surprisingly, this manuscript has removed information on sampling sites and source publications into the supplements. I checked the relevant documentation in the supplements, and it turned out to be very poor, actually completely unacceptable. Most problematically, no publication data are given (only titles or statements that look like titles, without any author, journal or publication date information; see Suppl. Table 1) and it is therefore impossible to tell with certainty whether the metagenomes used in this meta-analysis are already published. The map of sampling sites has low resolution; it does not even show the Amsterdam or Napoli mud volcanoes (singled out in Fig. 1) in the Mediterranean. Also, Supplementary figure 2 lists the samples only by acronym without any further information.

Given that any metagenomic meta-analysis would not even exist without its sources, it is essential to ask for a complete, transparent and thorough accounting of the source materials (sites, samples etc) and publications in the main manuscript. A good global map and a comprehensive table of the samples (with complete references!) are essential and need to be presented in full.

The title "Novel diazotrophs enhance productivity of cold seep sediment ecosystems in the deep sea" is classic overreach. There are no ecosystem productivity data in this manuscript. The claim of the title is simply not reflected in the manuscript content. What the manuscript shows is the phylogenetic diversity of diazotroph metagenomes in different cold seep sites. Everything else is extrapolation and inference. Please stick to what the manuscript actually shows, metagenome-derived diversity of diazotrophs in deep-sea cold seeps.

Details:

Lines 139 ff. Awkward phrasing, please revise for clarity.

Lines 192 and 193, and also line 202: The uncultured phyla UBA6262 and FCPU426 need a short explanation

Line 198: "...BuS5, Desulfatiglandaceae and Syntrophales." Please reference these alkane and aromatics-oxidizing sulfate-reducing lineages

Line 198: what is "this"? Presumably something like the increased diversity of diazotrophic lineages within the Desulfobacterota?

Line 213: What is MgATP hydrolysis?

Line 254 ff: What exactly is listed here? Metagenome-derived Genomes?

Line 344: Do sediments actually suffer? They are impacted by...

Line 354: "methane-dependent and independent metabolisms" is too vague. You mean methanogenesis (hydrogenotrophic or otherwise) and sulfate-dependent methane oxidation

The first figure should be a referenced global map of hydrocarbon seeps that were examined in this study. The annotation to this map could introduce the sample site acronyms that are used in subsequent figures, for example Figures 3 and 4.

Since different seep sites are shown color-coded in figure 3C, use the same color coding for the map.

Figure 1c. Does "Northern Gulf of Mexico seeps" refer to one particular seep or to a group of seeps? The global map (Suppl. Figure 1) shows three sampling sites in the Gulf of Mexico.

Figure 2. Does the color-coding of the labels have any particular meaning? Is there any connection with the color scheme in the phylogenetic trees (Figure 4a,b)? If so, please explain.

Figure 3a, please spell out the acronyms for the different sample types in a generally recognizable form. For example, the Haima and GoM samples are easy to identify, but where are the Amsterdam or Napoli mud volcano samples?

Figure 7. The numerical measures of nucleotide diversity and pN/pS ratios should be introduced briefly and contextualized for the benefit of the reader. If figure limits are a concern, this figure is the best candidate for moving into the supplements.

REVIEWER COMMENTS

Reviewer #1 (Remarks to the Author):

This study addresses an often overlooked metabolic feature of carbon-dominated environments like deep-sea hydrocarbon (oil and gas) seeps: the source of nitrogen for the seep associated microbial communities. Here, the authors analyzed the presence of nitrogen fixation genes in metagenomes from several cold seeps. A relatively high diversity of *nif* genes was found, including some novel clades. N₂ fixation appears to occur frequently in microorganisms involved in hydrocarbon oxidation, establishing a link between the C and N cycles at seeps sites.

This is an actual, much needed work. It reveals the diversity of N₂ fixers in deep sea habitats, the phylogenetic identity and energy metabolism of the harboring microorganisms. A criticism is that it is based purely on metagenomics analyses; transcriptomics, which is a logical step to demonstrate expression of potential function, was done for some of the sites, but not for all. In this respect, I believe some of the claims regarding novel nitrogenase genes (e.g. L 133) should be somewhat toned down (more in the specific comments): it is not clear if these are part of functional nitrogenases. Maybe more supporting arguments can be provided by the genetic context (Figure 4b), but the authors should bring this forward.

Response: We thank the reviewer for detailed and insightful comments. We agreed with the reviewer that transcriptomics is necessary to demonstrate *in situ* activities of cold seep diazotrophs. To address this concern, we added two additional metatranscriptomes from the Haima cold seep site. One is newly sequenced metatranscriptome (Haima5_0.3, 28-30 cmbsf) in this study, and another (S11_6, 160-170 cmbsf) is from our recent study (10.1101/2022.05.09.490922). The added metatranscriptomic analyses also demonstrated that *nifH* genes were actively transcribed at the Haima cold seep (see revised Supplementary Table 11). Note that sampling stations of Haima1, Haima2, Haima3 and the newly added Haima_S11 are actually all from the Haima cold seep. Therefore, we revised “13 globally distributed cold seeps” to “11 globally distributed cold seeps” in the manuscript. Additionally, we have toned down the claims regarding novel nitrogenase genes. Based on the results from genome-resolved analyses related to nitrogenase operon structure and conserved motif, MSL and group VII were considered to be potential functional nitrogenases in this study. For clarity, we noted this in the revised manuscript. Changes made:

Abstract (L28-30): “To fill this gap, we analyzed 61 metagenomes, 1428 metagenome-assembled genomes, and six metatranscriptomes derived from 11 globally distributed cold seeps.”

Introduction (L99-101): “To this end, gene- and genome-centric analyses of 61 metagenomes are coupled with six metatranscriptomes derived from 11 globally

distributed areas of hydrocarbon seepage (Figure 1 and Supplementary Table 1).”

Results and Discussion (L277-278): “... two metatranscriptomes sequenced from Haima cold seep sediments and four metatranscriptomes sequenced from Jiaolong cold seep sediments ...”

Result and Discussion (L280-283): “For both seep sites, the *nifH* genes of ANME-1, ETH-SRB1, and Caldatribacteriota were transcribed at moderate to high levels, up to 60-335 transcripts per million reads (TPM) (Supplementary Table 11), whereas fewer transcripts from ANME-2 were detected.”

Methods (L445-465): “One sediment layer from the Haima cold seep (Haima5_0.3, 28-30 cmbsf) was used for metatranscriptomic extraction and sequencing (Figure 1 and Supplementary Table 1). Total RNA was extracted from ~2.5 g of sediments using the RNeasy PowerSoil Total RNA Kit (Qiagen) according to the manufacturer’s instructions. Total RNA extracts were treated with DNase I (Vazyme, Nanjing, China) to remove DNA. RNA concentrations were evaluated on Qubit 2.0 Fluorometer (Invitrogen, Carlsbad, CA, USA). The quality of RNA was checked using gel electrophoresis. RNA was reverse transcribed to cDNA using Ovation Ultralow System V2 kit (NuGen, San Carlos, CA, USA). Metatranscriptomic libraries were prepared with the VAHTS Universal V8 RNA-seq Library Prep Kit (Vazyme, Nanjing, China) and sequenced on an Illumina NovaSeq 6000 platform. Four previously sequenced metatranscriptomes from Jiaolong (JL) cold seep sediments⁷¹ (JL_0.02, 0-2 cmbsf; JL_0.06, 4-6 cmbsf; JL_0.1, 8-10 cmbsf; JL_0.14, 12-14 cmbsf) and one metatranscriptome from the Haima cold seep (S11_6, 160-170 cmbsf) in our recent study¹⁰⁹ (Figure 1 and Supplementary Table 1) were also included in the metatranscriptomic analysis. All raw metatranscriptomic reads were quality controlled as described previously⁷¹. SortMeRNA v.4.3.4¹¹⁰ was used to remove rRNA reads. Metagenomic reads were mapped against *nifH*-containing MAGs, with those having >5× coverage at Haima and JL cold seep sites retained for further transcript per million (TPM) calculations. The TPM of metatranscriptomic clean reads mapped to predicted genes from *nifH*-containing MAGs were calculated using Salmon v0.13.1 (-meta -validateMappings)¹¹¹.”

Result and Discussion subtitle (L110): “Cold seeps harbour canonical and novel nitrogenase gene homologues”.

Result and Discussion (L134-137): “... including (1) a clade similar to *nifH* found in *Methanosarcina* species but not clearly falling into the canonical groups (i.e., *Methanosarcina*-like group, MSL; n=7), (2) a novel clade proposed here as group VII (n=15), and (3) a novel clade proposed here as group VIII containing *nifH*-like genes (n=6).”

Result and Discussion (L137-139): “Among the three novel lineages, MSL and group VII were considered as bona fide *nifH* based on the analyses of nitrogenase operon

structure and conserved motif detailed below.”

A second major issue is the first chapter of the Results and Discussion and Figure 1: this is not original work, it is a compilation of former publications, with additional statistical analyses – in a sense what one could expect to find in a review manuscript. I understand that in the absence of direct measurements this is meant to show that N₂ fixation is generally an important process at seep sites – this is fine; however, I’m not convinced the R & D section should start with a literature overview: it is in part misleading, since it leaves the impression of original data (for example, much of the Figure 1 legend does not hint that this is a literature summary). In my opinion, this information (including citations of the original papers) belongs to the Introduction as a background and motivation for the current work.

A second argument for the above is that Figure 1 shows ¹⁵N isotope data of sites which were not part of the current metagenome work, like the Napoli and Amsterdam mud volcanoes (if one compares the Supplementary Tables 1 and 2). Including it as part of the present study leaves the impression of disconnected parts: they can hardly be directly correlated to the metagenome work, since they come from other sites or other sampling events.

Response: We fully agree with this comment. As also suggested by the reviewer in the comment below, we have integrated the first chapter of Results and Discussion into Introduction and moved the ¹⁵N isotope data into Supplementary Information. To avoid misleading, we emphasized that ¹⁵N data is a literature summary (see Supplementary text). In addition, the statement about ¹⁵N data in Abstract was removed. Changes made:

Figure 1 now is Supplementary Figure 1.

Introduction (L101-105): “Samples originate from five types of cold seeps, namely gas hydrates, mud volcanoes, asphalt volcanoes, oil and gas seeps and methane seeps. Most seep types have previously been shown to have lighter $\delta^{15}\text{N}$ indicative of biological nitrogen fixation, compared to nearby background sediment^{13, 41, 42} (detailed in Supplementary Information).”

Supplementary Information: Descriptions for the compilation of $\delta^{15}\text{N}$ records of bulk sediment organic matter from published literature

L 43-46: I suggest to split this sentence for clarity

Response: Split as suggested. Changes made:

Introduction (L43-46): “Cold seeps are often classified as slow-flow mineral-prone or high-flux mud-prone systems according to their hydrocarbon fluid regime¹. They span oil and gas seeps, methane seeps, gas hydrates, asphalt volcanoes, mud volcanoes, brine

pools, and brine basins among others.”

L 52-54: One of the initial studies showing alkane oxidation at seep sites is Kleindienst et al., ISME J 2014, 8:2029-2044

Response: We thank the reviewer for this comment. We now added this important reference here. See reference 12.

L 91: I believe that ‘high-energy electrons’ has another meaning than the context here. Perhaps high-potential electrons?

Response: We corrected this phrase as “high-potential electrons”.

L 102-105: citations are needed here; the first section of R & D can be integrated here

Response: We added relative citations here. Additionally, we integrated here the first subsection of Results and Discussion. Changes made:

Figure 1 now is Supplementary Figure 1.

Introduction (L101-105): “Samples originate from five types of cold seeps, namely gas hydrates, mud volcanoes, asphalt volcanoes, oil and gas seeps and methane seeps. Most seep types have previously been shown to have lighter $\delta^{15}\text{N}$ indicative of biological nitrogen fixation, compared to nearby background sediment^{13, 41, 42} (detailed in Supplementary Information).”

L 110: what is ‘purifying selection’?

Response: Natural selection includes three types, i.e. positive selection, balancing selection and purifying selection or negative selection. Negative selection or purifying selection is the selective removal of alleles and indels (insertions or deletions) that cause truncation (particularly malfunction-inducing) of the original functional gene. As requested by Reviewer #2, we revised the last paragraph of Introduction and this sentence was removed accordingly in the revised version. We now added an explanation for purifying selection in the revised text in Result and Discussion. Changes made:

Result and Discussion (L330-333): “The ratios of the two rates of non-synonymous to synonymous polymorphism (pN/pS) in *nifHDK* were determined (Figure 7b) to assess if genes are under purifying (negative) selection which involves the selective removal of deleterious mutations⁷⁹.”

L 133: I don’t think phylogenetic clustering alone can be used to give a verdict if these genes encode nitrogenases or not. Many of the genes retrieved here are nifH homologues (according to L 150)

Response: We agree and toned down the language here. Now the subtitle was changed as follows: “Cold seeps harbour canonical and novel nitrogenase gene homologues”.

L 143: facultative (?) bacteria?; same line, delete ‘anaerobic’ in front of ‘Mo-Fe nitrogenases’.

Response: We thank the reviewer for pointing this out. We modified “facultative bacteria” to “facultative anaerobic bacteria” and deleted “anaerobic” in front of “Mo-Fe nitrogenases” as suggested. Changes made:

Result and Discussion (L118-120): “These include (1) typical Mo-Fe nitrogenases from aerobic and facultative anaerobic bacteria (group I; n=1); (2) Mo-Fe nitrogenases from anaerobic bacteria and archaea (group II; n=32) ...”

L 158-161: strictly speaking, nitrogenases are enzyme complexes, composed of several subunits, NifH being one of them. According to Figure 2, these are NifH or NifH-homologues clusters, not nitrogenase clusters.

Response: We agree with the reviewer and rephrased this sentence to better clarify this. Changes made:

Result and Discussion (L134-137): “... including (1) a clade similar to *nifH* found in *Methanosarcina* species but not clearly falling into the canonical groups (i.e., *Methanosarcina*-like group, MSL; n=7), (2) a novel clade proposed here as group VII (n=15), and (3) a novel clade proposed here as group VIII containing *nifH*-like genes (n=6).”

L 176-178: the first paper showing Mcr involvement in multi-carbon alkane oxidation is Laso-Perez et al., Nature 2016, 539:396-401

Response: We thank the reviewer for this comment. We now added this important reference here. See reference 55.

L 191 and elsewhere in the manuscript refers to Halobacteriota, which is shown in Figure 4a to include Methanosarcina and ANME; according to NCBI Taxonomy, Methanosarcina are part of Methanomicrobia, not Halobacteriota. Am I maybe missing something some recent revision of taxonomy?

Response: In this manuscript, we used the Genome Taxonomy Database (GTDB) taxonomy (<https://gtdb.ecogenomic.org/>). In the GTDB system, Methanosarcinia is the name of one class, which belongs to the phylum of Halobacteriota: d__Archaea; p__Halobacteriota; c__Methanosarcinia; o__Methanosarcinales; f__Methanosarcinaceae; g__Methanolobus; s__Methanolobus psychrophilus.

However, we realize that some readers might also prefer the NCBI system. To facilitate

comparison with previous publications, we also added the corresponding names in NCBI taxonomy. Changes made can be found in the whole subsection of “Diverse diazotrophs from ten different phyla reside in cold seep sediments” and other places in this manuscript.

L 201: genomic evidence of nitrogen fixation: since this is based on gene presence only, it should be slightly toned down. In the absence of process demonstration, presence of genes indicates potential.

Response: We corrected the sentence to tone down the language. Changes made:

Result and Discussion (L185-188): “Indeed, this represents the first genomic evidence of nitrogen fixation potential in five different phyla, namely Altarchaeia, Omnitrophota, Caldatribacteriota along with two bacterial candidate phyla FCPU426 and UBA6262^{48, 58}.”

L 208: replace nifH (gene) with NifH (protein), in accordance with Fig 4b

Response: Changed as suggested.

L 238 and forward: I am not familiar with the use of the term ‘recruitment’ in this context. Does it refer to relative abundance?

Response: Yes, it referred to relative abundance. We changed this term to be clear. Changes made:

Result and Discussion (L226-229): “When considered individually, Desulfobacterota (comprising up to 2% of the microbial community) and Caldatribacteriota (also up to 2%) represented the major bacterial diazotrophs, and Halobacteriota constituted major archaeal diazotrophs (up to 13%).”

L 258 and elsewhere: should refer to BuS5 as strain BuS5 or Desulfosarcina sp. strain BuS5

Response: In this manuscript, we used the Genome Taxonomy Database (GTDB) taxonomy (<https://gtdb.ecogenomic.org/>). In the GTDB system, BuS5 is the name of one genus, which belongs to the phylum of Desulfobacterota. Desulfosarcina sp. BuS5 belongs to the current GTDB taxonomy: d__Bacteria; p__Desulfobacterota; c__Desulfobacteria; o__Desulfobacterales; f__BuS5; g__BuS5; s__BuS5 sp000472805. To facilitate comparison with previous publications, we also added the corresponding names in NCBI taxonomy. Changes made:

Result and Discussion (L178-181): “Within the Desulfobacterota, nitrogenase-encoding MAGs belonged to the order of “C00003060” (aka SEEP-SRB1c³⁹) along with other non-ANME-associated bacterial groups such as BuS5 (aka *Desulfosarcina*

sp. BuS5 in NCBI taxonomy), *Desulfatiglandaceae* and *Syntrophales* ...”

Section starting on L 275: without reference to works demonstrating that H₂ production during N₂ fixation can be quantitatively relevant in as much as to have an impact on the overall metabolism of other microorganisms, I find this whole section rather speculative; it is not supported by data obtained in this study: the cited table (Supplementary Table 10) is a list of H₂ases, does not show quantitative production of H₂ during N₂ fixation.

Response: Thank you for this comment. This sentence is indeed confusing. We removed the description of “large amounts of hydrogen production”. Additionally, we added context and relative references showing that nitrogenases catalyze the production of H₂ as a by-product of nitrogen reduction to ammonia. In this way, we only emphasize that nitrogen fixation can lead to hydrogen production. Changes made:

Result and Discussion (L264-269): “Nitrogenases not only mediate the reduction of molecular nitrogen into ammonia, but also reduce protons into molecular hydrogen during their reaction cycle⁶⁹. Some diazotrophs identified here, including those within Caldatribacteriota, Desulfobacterota and *Methanosarcinaceae* (Supplementary Table 10), have the potential to internally recycle this hydrogen as an energy source, for example by using group 1 [NiFe]-hydrogenases linked to anaerobic respiratory chains⁷⁰.”

L 339 and forward: still not clear what ‘purifying selection’ means

Response: See response to the previous comments on the same point. Changes made:

Result and Discussion (L330-333): “The ratios of the two rates of non-synonymous to synonymous polymorphism (pN/pS) in *nifHDK* were determined (Figure 7b) to assess if genes are under purifying (negative) selection which involves the selective removal of deleterious mutations⁷⁹.”

The Conclusion section brings little new, overarching interpretation of the data, and it reads more like a rehearsal of previous statements. The sentence at L 353, ‘These diazotrophs regularly transcribe nifH ...’ is not really supported by the data presented here. I am wondering if the authors could comment on the apparent correlation between the hydrocarbon-derived C and N cycles, since 23 out of the 35 diazotrophic MAGs recovered (if I was correctly reading Figure 5) are of microorganisms involved directly or indirectly in hydrocarbon oxidation. Could one speculate that selective pressure was ‘pushing’ N₂ fixation genes into the microorganisms making use of the most abundant energy source at seeps?

Response: We agree with this comment. We revised Conclusion as suggested by the reviewer. Changes made:

Conclusion (L341-343): “In the deep sea cold seep sediments that are impacted by darkness, low temperatures, and high hydrostatic pressure, growth of microbiomes consuming rich hydrocarbons is also supposed to be nitrogen limited.”

Conclusion (L350-355): “Of the 35 recovered diazotrophic MAGs, 23 represent microorganisms that are involved directly or indirectly in hydrocarbon metabolisms, including anaerobic methane oxidizing archaea and anaerobic non-methane alkane-degrading bacteria. The tight correlation between hydrocarbon-derived carbon and nitrogen cycles indicates that nitrogen fixation pathways might be selected for microorganisms making use of the most abundant energy source at cold seeps.”

Figure 4: it would be helpful for the reader if panels a and b are correlated by using similar colors for the same groups.

Response: We thank the reviewer for this suggestion. We corrected Figure 4 to make same groups with the same colors (see revised Figure 4).

Figure 6: maybe I missed that, but it seems this figure is not cited in the text

Response: Sorry for this mistake. We now cited Figure 6 in the main text.

Reviewer #2 (Remarks to the Author):

In the manuscript entitled “Novel diazotrophs enhance productivity of cold seep sediment ecosystems in the deep sea”, Xiyang Dong, Chuwen Zhang and colleagues analyzed 61 metagenomes and fewer metatranscriptomes from 13 globally distributed deep-sea cold seeps in order to expand our understanding of nitrogen fixation by various diazotrophic taxa in these environments. The authors assessed the phylogeny of nifH genes from their metagenomic assemblies (gene-centric) and subsequently characterized a considerable number of environmental genomes (genome-resolved metagenomics), 35 of which contain nitrogenase genes. Authors demonstrated with genomics that a series of phyla are capable of performing nitrogen fixation, an important contribution to the field. They further explored the metabolic capabilities of those diazotrophs to understand how they acquire energy needed for nitrogen fixation. I particularly appreciated information presented in the figure 4 and more broadly the considerable efforts made by the authors to convey their insightful results throughout the manuscript. Finally, I note that key data produced in this study (includes the assemblies and MAGs) have been made public.

Response: We thank the reviewer for the careful reading and positive comments.

I did not identify any potential flaw. I have two comments (the second is optional)

that might be considered to improve the strength of the study, followed by a series of smaller points.

Comment one: The manuscript is well written, and the introduction clearly describes our current understanding of nitrogen fixation in cold seeps. However, the last paragraph of the introduction is a mix of introduction (first sentence), results (description of the samples) and discussion (summary of results). In my view, authors should remove results and discussion from this section, and simply describe briefly that they studied publically available metagenomes and metatranscriptomes from various seeps to better understand taxa contributing to nitrogen fixation in these ecosystems using gene and genome-centric analyses coupled with $\delta^{15}\text{N}$ records of bulk sediment organic matter.

Response: As suggested, we removed results and discussion from Introduction. Also as suggested by Reviewer #1, we integrated the first section of the Result and Discussion from the previous version into this section. Changes made:

Introduction (L97-108): “In this study, we investigate the hidden diversity and distributions of nitrogenases and diazotrophs, and compile evidence for their *in situ* activities within deep-sea cold seep sediments. To this end, gene- and genome-centric analyses of 61 metagenomes are coupled with six metatranscriptomes derived from 11 globally distributed areas of hydrocarbon seepage (Figure 1 and Supplementary Table 1). Samples originate from five types of cold seeps, namely gas hydrates, mud volcanoes, asphalt volcanoes, oil and gas seeps and methane seeps. Most seep types have previously been shown to have lighter $\delta^{15}\text{N}$ indicative of biological nitrogen fixation, compared to nearby background sediment^{13, 41, 42} (detailed in Supplementary Information). Overall, this study corroborates deep-sea cold seep sediments as overlooked habitats for uncovering diverse diazotrophs from uncultivated lineages supported by diverse energy sources, and emphasizes the importance of nitrogen fixation in a carbon-dominated environment.”

Comment two: Authors performed automatic binning on metagenomic assemblies and co-assemblies without any visualization and curation of especially the diazotroph MAGs. As an extra-step used in similar studies but not here, the 35 diazotroph MAGs could be imported into a platform with enhanced visualization capabilities dedicated to genome-resolved metagenomics for curation purposes, thus enhancing confidence in the quality of MAGs. While optional given the current status of our standards, providing a database of 35 manually curated diazotrophic MAGs for cold seeps could have been viewed as a better reference in years to come. At times, errors of automatic binning can be surprising...

Response: We fully agree with this comment. Automatic binning can cause artifacts. We did not use visual platforms to curate these MAGs. Instead, we used a recently developed entropy-based approach GUNC (<https://grp-bork.embl-community.io/gunc/>) to detect the chimerism and contamination of the 35 diazotrophic MAGs. The results

showed that 29 of 35 diazotrophic MAGs passed GUNC filters and had no chimerism (see revised Supplementary Table 5). For the other six MAGs, they are poorly represented in the reference and therefore it is difficult for GUNC to judge if they are chimeric. Changes made:

Methods (L389-390): “Additionally, we used GUNC v1.0.1⁹⁴ to assess chimerism and contamination of the diazotrophic MAGs.”

Smaller points:

Ln 70-74: References regarding the nitrogen fixation by Cyanobacteria and heterotrophic bacteria in the Surface Ocean and deeper layers of the oceans (introduction) might be improved to better reflect the literature. Authors should make sure references for the deeper layers only cover studies for the deeper layers (I could see at least one problem there).

Response: Thank you for this comment. We carefully checked these references and rearranged them to better reflect the literature. Change made:

Introduction (L71-74): “Multiple cyanobacterial diazotrophs are responsible for a substantial portion of new nitrogen input in the surface ocean²¹⁻²⁴. Various diazotrophs are also active in both surface and deeper waters, including diverse heterotrophic Proteobacteria²⁵⁻²⁸.”

Ln 162: Number of nifH genes considered bona fide is required here to better understand the methodology.

Response: We thank the reviewer for this comment. We added the number here (n=66).

Ln 165: Should be “total bacterial community”, or “total bacterial and archaeal community” depending on the single copy core genes used. This is assuming that eukaryotes and viruses are excluded from the analysis.

Response: We changed it to “total bacterial and archaeal community” to be clear.

Ln 185: It is ok to use newly defined phylum named by GTDB but it should be stated that it is GTDB taxonomy and not the commonly used NCBI taxonomy (e.g., Planctomycetota for planctomycetes).

Response: We agree with this comment. We now added the note that we used GTDB taxonomy. Additionally, we also added the corresponding names in NCBI taxonomy for each phylum for the reason of literature comparison. Changes made:

Result and Discussion (L161-170): “Using metagenomic assembly and binning strategies, we recovered 1428 non-redundant bacterial (n=1146) and archaeal (n=282) population genomes (Supplementary Table 4) belonging to 76 phyla based on the

Genome Taxonomy Database (GTDB; see Methods). Most genomes were affiliated with the phyla Chloroflexota (n=239, namely Chloroflexi in NCBI taxonomy), Desulfobacterota (n=185, namely Deltaproteobacteria), Halobacteriota (n=114, namely Euryarchaeota), Proteobacteria (n=130), Acidobacteriota (n=70, namely Acidobacteria), Bacteroidota (n=65, namely Bacteroidetes), Planctomycetota (n=54, namely Planctomycetes), Thermoplasmata (n=48, namely Thermoplasmata), and Asgardarchaeota (n=43, namely Asgard superphylum).”

Result and Discussion (L172-176): “... belong to the Halobacteriota (n=14), Desulfobacterota (n=11), Chloroflexota (n=2), UBA6262 (n=2, candidate phylum), Altarchaeota (n=1), Caldatribacteriota (n=1, namely Atribacteria), Omnitrophota (n=1, namely Omnitrophica), FCP426 (n=1, candidate phylum), Verrucomicrobiota (n=1, namely Verrucomicrobia) ...”

Result and Discussion (L180-181): “... BuS5 (aka *Desulfosarcina* sp. BuS5 in NCBI taxonomy) ...”

Ln 189: 35 out of >1300 MAGs contain nitrogenase genes. Are those the most abundant ones, or is that in clear contrast with the estimate of diazotroph proportion within the community as assessed from *nifH* genes and single copy core genes? Needed would be the proportion of mapped reads for the 35 diazotrophic MAGs compared to all MAGs across metagenomes. If the proportion is far below 20% then there is incoherence that should be discussed. However, I agree with the authors that both the *nifH* and MAG analyses emphasise that some diazotrophs are very abundant in those systems, hence the relevance of such study.

Response: Based on read abundance ratios of bona fide *nifH* and single copy ribosomal protein genes, diazotrophs were at $24 \pm 22\%$ of the total community. However, based on read mapping to the 35 diazotrophic MAGs, their total relative abundance is $4 \pm 3\%$, much lower than the estimation from *nifH* genes. This suggests that there are still some diazotrophic MAGs that we did not recover here. Also, some diazotrophs might contain multiple copies of *nifH*. We added this discussion in the main text. Changes made:

Result and Discussion (L222-226): “Their overall relative abundance is $4 \pm 3\%$, far below the estimated values based on read abundance ratios of *nifH* genes (Supplementary Table 3). Two possible explanations might account for this: (1) diazotrophic MAGs might contain multiple copies of *nifH*; (2) there are still some diazotrophs that we did not recover here.”

Ln 310: Figure 6 is not referred to in the main text. It could be cited here.

Response: Thank you. We have cited Figure 6 here.

Ln 333: it is unclear what the authors refer to with accumulation of mutations.

Response: We corrected this sentence to be clear. Changes made:

Result and Discussion (L329-330): “Nucleotide diversity was also estimated to be similar among *nifH*, *nifD* and *nifK* genes (Figure 7a).”

Ln 356: impact of HGTs is not demonstrated. Authors should modify their statement. The genes simply co-occur in the same genomic regions. If authors want to claim HGT, they need to show incoherence between the phylogenomics of genomes and the phylogeny of *nifH* genes. As far as I could see from the figure 4, the coherence between the two is quite important. I would like a further explanation by the authors on this matter.

Response: Thanks for this comment. We agree that the incoherence between gene tree and genome tree is necessary for HGT. To address this concern, we have added detailed discussions on the impact of HGTs among seep diazotrophic groups in the section of Result and Discussion.

Result and Discussion (L312-318): “Meanwhile, the phylogenies of most NifH sequences were observed to be inconsistent with their corresponding taxonomies (Figure 4). Except for *Methanosarcina*-like group NifH, sequences from group II, group III and group VII were scattered among diverse bacterial and archaeal phyla (Figure 4). For example, six different bacterial phyla encoded NifH sequences of group II. Combined with the MGEs analysis, these results suggest that HGTs occurred among cold seep communities during their evolution.”

Reviewer #3 (Remarks to the Author):

This is a meta-analysis of published metagenomes for diazotrophy-relevant genes, supplemented with some geochemical information (N stable isotopes) on the sampling sites. Surprisingly, this manuscript has removed information on sampling sites and source publications into the supplements. I checked the relevant documentation in the supplements, and it turned out to be very poor, actually completely unacceptable. Most problematically, no publication data are given (only titles or statements that look like titles, without any author, journal or publication date information; see Suppl. Table 1) and it is therefore impossible to tell with certainty whether the metagenomes used in this meta-analysis are already published. The map of sampling sites has low resolution; it does not even show the Amsterdam or Napoli mud volcanoes (singled out in Fig. 1) in the Mediterranean. Also, Supplementary figure 2 lists the samples only by acronym without any further information.

Response: We apologize for the ambiguity here. The ¹⁵N isotope data presented in the old Figure 1 were compiled from previous publications, aiming to demonstrate the

occurrence of nitrogen fixation at cold seep ecosystems from the aspect of geochemistry. The sites for ^{15}N isotope measurements including Amsterdam or Napoli mud volcanoes, Northern Gulf of Mexico along with Site F and Haima methane seeps did not match metagenomics analyses. To avoid confusion, we integrated the first section of Result and Discussion into the Introduction section and Supplementary Information, as suggested by Reviewer #1. The old Figure 1 now is Supplementary Figure 1, and the map has been redrawn and now is Figure 1, in line with the comment on the same issue below.

Metagenomes and metatranscriptomes used in this study were compiled or produced from sediment samples collected from 11 geographically diverse cold seep sites. We have provided detailed information for these data in revised Supplementary Table 1.

Most of the data were produced by us and described in detail in our previous publications, including a Scotian Basin cold seep in the northwest Atlantic Ocean (10.1038/s41467-020-19648-2), and the South China Sea cold seeps Jiaolong, Haiyang4, Site F and Haima (10.1007/s42995-020-00057-9, 10.1111/1462-2920.15796, 10.1016/j.dsr.2021.103489, and 10.1101/2022.05.09.490922). To facilitate the data access, we further deposited the metagenomic and metatranscriptomic datasets from the South China Sea cold seeps in NCBI BioProject databases (PRJNA831433, <https://dataview.ncbi.nlm.nih.gov/object/PRJNA831433?reviewer=m8adl5stluh8er8100l69213i1>).

Other metagenomic data were downloaded from NCBI's Sequence Read Archive, including datasets derived from Haakon Mosby mud volcano, Eastern North Pacific ODP site 1244, Mediterranean Sea Amon mud volcano, Santa Monica Mounds, and Gulf of Mexico. These data all have been used in other previous publications including ours (e.g. 10.1038/s41564-018-0171-1, 10.1128/msphere.00785-21, 10.1038/s41396-021-00932-y and 10.1101/2022.05.09.490922).

We also acknowledged the producers of external data that are not from us in the section of "Acknowledgements". See Acknowledgements: "The study would not have been possible without publicly available genomic data from already published studies, and the authors wish to acknowledge all the participants, organizations, and funding agencies that contributed to the collection and analysis of all the data utilized in our study, including those acknowledged in Glass et al. 2021, Laso-Pérez et al. 2019, Ruff et al. 2018, Zhao et al. 2020, and Yu et al. 2018. Detailed references are listed in Supplementary Table 1."

Given that any metagenomic meta-analysis would not even exist without its sources, it is essential to ask for a complete, transparent and thorough accounting of the source materials (sites, samples etc) and publications in the main manuscript. A good global map and a comprehensive table of the samples (with complete references!) are essential and need to be presented in full.

Response: We thank the reviewer for this comment. We have redrawn the global map and revised the Supplementary Table 1 to provide comprehensive information on samples used in our meta-analysis. See the response to last comment.

The title "Novel diazotrophs enhance productivity of cold seep sediment ecosystems in the deep sea" is classic overreach. There are no ecosystem productivity data in this manuscript. The claim of the title is simply not reflected in the manuscript content. What the manuscript shows is the phylogenetic diversity of diazotroph metagenomes in different cold seep sites. Everything else is extrapolation and inference. Please stick to what the manuscript actually shows, metagenome-derived diversity of diazotrophs in deep-sea cold seeps.

Response: We thank the reviewer for this comment. The title was changed as follows: Phylogenetically novel and catabolically diverse diazotrophs reside in deep-sea cold seep sediments.

Details:

Lines 139 ff. Awkward phrasing, please revise for clarity.

Response: We revised this sentence. Changes made:

Result and Discussion (L116-118): "The phylogenetic tree (Figure 2) suggested that *nifH* homologues were classified into distinct bona fide nitrogenase sequences (canonical groups I to III) as well as nitrogenase-like groups (groups IV to VI)⁴⁴⁻⁴⁸."

Lines 192 and 193, and also line 202: The uncultured phyla UBA6262 and FCPU426 need a short explanation

Response: The taxonomy in this manuscript was based on GTDB R06-RS202. Phyla UBA6262 and FCPU426 are still candidate names so far. As suggested, we added a short explanation for them in the revised text. Changes made:

Result and Discussion (L172-175): "... belong to the Halobacteriota (n=14), Desulfobacterota (n=11), Chloroflexota (n=2), UBA6262 (n=2, candidate phylum), Altarchaeota (n=1), Caldatribacteriota (n=1, namely Atribacteria), Omnitrophota (n=1, namely Omnitrophica), FCPU426 (n=1, candidate phylum) ..."

Result and Discussion (L185-188): "Indeed, this represents the first genomic evidence of nitrogen fixation potential in five different phyla, namely Altarchaeia, Omnitrophota, Caldatribacteriota along with two bacterial candidate phyla FCPU426 and UBA6262^{48, 58}."

Line 198: "...BuS5, Desulfatiglandaceae and Syntrophales." Please reference these alkane and aromatics-oxidizing sulfate-reducing lineages

Response: We added an explanation and references here for these lineages. Changes made:

Result and Discussion (L180-182): "... BuS5 (aka *Desulfosarcina* sp. BuS5 in NCBI taxonomy), *Desulfatiglandaceae* and *Syntrophales* known to degrade alkanes or aromatic hydrocarbons coupled with sulfate reduction^{7,57}."

Line 198: what is "this"? Presumably something like the increased diversity of diazotrophic lineages within the Desulfobacterota?

Response: Thanks for this comment. We revised this sentence to make it clear. Changes made:

Result and Discussion (L183-185): "The increased diversity of bacterial and archaeal diazotrophic lineages substantially broadens the genomic database of microbial diazotrophs in deep sea cold seep sediments ..."

Line 213: What is MgATP hydrolysis?

Response: MgATP hydrolysis is equal to ATP hydrolysis. Mg²⁺ is an important cation for maintaining cellular functions. In living cells, ATP binds to Mg²⁺ in order to be biologically active, forming MgATP complexes. For clarity, we changed "MgATP hydrolysis" to "ATP hydrolysis" in the revised text.

Line 254 ff: What exactly is listed here? Metagenome-derived Genomes?

Response: Yes, they are MAGs. We noted this in the revised manuscript. Changes made:

Result and Discussion (L241-242): "Genomic analyses of these 35 MAGs identified four distinct groups regarding carbon cycling ..."

Line 344: Do sediments actually suffer? They are impacted by...

Response: We changed "suffer from" to "impacted by" to be accurate.

Line 354: "methane-dependent and independent metabolisms" is too vague. You mean methanogenesis (hydrogenotrophic or otherwise) and sulfate-dependent methane oxidation

Response: As requested by Reviewer #1, we revised the section of Conclusion and this sentence was removed in the revised version.

The first figure should be a referenced global map of hydrocarbon seeps that were examined in this study. The annotation to this map could introduce the sample site acronyms that are used in subsequent figures, for example Figures 3 and 4.

Response: According to the comment from Reviewer #1, we now moved old Figure 1 to Supplementary Information (see revised supplementary Figure 1). We revised the old supplementary Figure 1 as Figure 1, which also included the sample site acronyms as suggested.

See Figure 1 and its legend (L809-815): “Geographic distribution of 11 cold seep sites where metagenomic and metatranscriptomic data were collected. These samples were originated from five types of cold seeps: gas hydrates, mud volcanoes, asphalt volcanoes, oil and gas seeps, methane seeps. Sites with red asterisks denote that both metagenomes and metatranscriptomes were collected, sites with blue asterisks denote that only metatranscriptomes were collected, and sites without asterisks denote that only metagenomes were collected. Also see details in Supplementary Table 1.”

Since different seep sites are shown color-coded in figure 3C, use the same color coding for the map.

Response: As suggested, we used the same color coding as Figure 3C for the map. See revised Figure 1.

Figure 1c. Does “Northern Gulf of Mexico seeps” refer to one particular seep or to a group of seeps? The global map (Suppl. Figure 1) shows three sampling sites in the Gulf of Mexico.

Response: We apologize for this confusion here. They are different sampling sites. The seep sites for geochemical data did not match with the ones for metagenomic analyses in the main text. We actioned the suggestion from Reviewer #1 and now integrated the statement related to geochemical data into the section of Introduction and also the Supplementary Information. See revised Figure 1 for the details of the sampling sites used for metagenomes and metatranscriptomes.

Figure 2. Does the color-coding of the labels have any particular meaning ? Is there any connection with the color scheme in the phylogenetic trees (Figure 4a,b) ? If so, please explain.

Response: The color-coding of the labels represents different groups of *nif* gene homologues. We now corrected Figure 2 to make it with the same color scheme as shown in Figures 3a and 4. See revised Figure 2.

Figure 3a, please spell out the acronyms for the different sample types in a generally recognizable form. For example, the Haima and GoM samples are easy to identify, but where are the Amsterdam or Napoli mud volcano samples?

Response: Thanks for this suggestion. The sample site acronyms in Figure 3 are the same as those in the global map (see revised Figure 1). The Amsterdam/Napoli mud volcano samples and other samples in the geochemical data did not connect to the

metagenomics work. We actioned the suggestion from Reviewer #1 and now integrated this part into the section of Introduction and also the Supplementary information. Changes made:

Figure 3 legend (L831): “The abbreviations of the sites are shown in Figure 1.”

Figure 7. The numerical measures of nucleotide diversity and pN/pS ratios should be introduced briefly and contextualized for the benefit of the reader. If figure limits are a concern, this figure is the best candidate for moving into the supplements.

Response: Thanks for this suggestion. We added explanations for nucleotide diversity and pN/pS ratios in the corresponding figure legend. Changes made:

Figure 7 legend (L850-856): “Evolutionary metrics of nitrogen fixation genes. (a) Nucleotide diversity (π) of *nifHDK* genes at different types of cold seeps; (b) pN/pS ratio of *nifHDK* genes at different types of cold seeps. Nucleotide diversity is used to measure genetic diversity within a population (microdiversity), which is calculated using the formula: $1 - [(\text{frequency of A})^2 + (\text{frequency of C})^2 + (\text{frequency of G})^2 + (\text{frequency of T})^2]$. pN/pS is the ratio of non-synonymous to synonymous polymorphism rates within a population.

Reviewer comments, second round -

Reviewer #1 (Remarks to the Author):

I believe that the manuscript has been substantially revised. My comments and suggestions have been considered or answered in full. I think publication in Nature Communications is now justified.

A few remaining comments:

Fig. 5a, Fig. 5c, formyl-H4MPT is misspelled; Fig. 5d: Beta-oxidation

Fig. 5d: at least *D'sarcina* BuS5 will not use the *Acd* for substrate level phosphorylation, but rather ATPases (<https://doi.org/10.1111/1462-2920.15956>). These strains are complete oxidizers, converting their substrate to CO₂. As depicted here, the pathway suggests incomplete oxidation of hydrocarbons to acetate. A suggestion is to depict the pathway branching at acetyl-CoA, one branch maintained as is now, another indicating conversion to CO₂.

Fig. 6: these are genes, I believe it is more appropriate to use the gene instead of protein indicatives: *nifH* (italics) instead of NifH, etc.

Reviewer #2 (Remarks to the Author):

The authors have appropriately considered all of my comments. I have no further points to make regarding this important study of new lineages of diazotrophs.

(for future studies, I simply would like the authors to seriously consider curation of at least the focal MAGs with *anvi'o*. GUNC or similar automated tools have very limited capabilities in comparison.)

Tom Delmont

Reviewer #3 (Remarks to the Author):

The revision has addressed my comments satisfactorily, and the manuscript is now much improved and more coherent.

A detail:

Supplementary Tables 1 and 2: In the column "Source Reference", please provide the full references, not just the titles.

In Suppl. Table 1, it makes no sense to list authors, journal and publication date in separate columns; all this should be included in the source reference.

REVIEWER COMMENTS

Reviewer #1 (Remarks to the Author):

I believe that the manuscript has been substantially revised. My comments and suggestions have been considered or answered in full. I think publication in Nature Communications is now justified. A few remaining comments:

Fig. 5a, Fig. 5c, formyl-H4MPT is misspelled; Fig. 5d: Beta-oxidation

Response: We thank the reviewer for reading our paper carefully. We have now corrected Fig.5a, Fig. 5c and Fig. 5d as suggested.

Fig. 5d: at least D'sarcina BuS5 will not use the Acd for substrate level phosphorylation, but rather ATPases (<https://doi.org/10.1111/1462-2920.15956>). These strains are complete oxidizers, converting their substrate to CO₂. As depicted here, the pathway suggests incomplete oxidation of hydrocarbons to acetate. A suggestion is to depict the pathway branching at acetyl-CoA, one branch maintained as is now, another indicating conversion to CO₂.

Response: We thank the reviewer for pointing out. Done as suggested.

Fig. 6: these are genes, I believe it is more appropriate to use the gene instead of protein indicatives: *nifH* (italics) instead of NifH, etc.

Response: We agree with the reviewer and have now replaced protein indicatives with genes.

Reviewer #2 (Remarks to the Author):

The authors have appropriately considered all of my comments. I have no further points to make regarding this important study of new lineages of diazotrophs. (for future studies, I simply would like the authors to seriously consider curation of at least the focal MAGs with *anvi'o*. GUNC or similar automated tools have very limited capabilities in comparison.)

Response: Thank you for this suggestion. We will fully consider curation of MAGs with *anvi'o* in our future studies.

Reviewer #3 (Remarks to the Author):

The revision has addressed my comments satisfactorily, and the manuscript is now

much improved and more coherent.

A detail:

Supplementary Tables 1 and 2: In the column “Source Reference”, please provide the full references, not just the titles. In Suppl. Table 1, it makes no sense to list authors, journal and publication date in separate columns; all this should be included in the source reference.

Response: We thank the reviewer for this comment and have now provided the full references as suggested (now as Supplementary Data 1 and 11).